# Semantic Score Distillation Sampling for Compositional Text-to-3D Generation

## Abstract

Generating high-quality 3D assets from textual descriptions remains a pivotal challenge in computer graphics and vision research. Due to the scarcity of 3D data, state-of-the-art approaches utilize pre-trained 2D diffusion priors, optimized through Score Distillation Sampling (SDS). Despite progress, crafting complex 3D scenes featuring multiple objects or intricate interactions is still difficult. To tackle this, recent methods have incorporated box or layout guidance. However, these layout-guided compositional methods often struggle to provide fine-grained control, as they are generally coarse and lack expressiveness. To overcome these challenges, we introduce a novel SDS approach, Semantic Score Distillation Sampling (SemanticSDS), designed to effectively improve the expressiveness and accuracy of compositional text-to-3D generation. Our approach integrates new semantic embeddings that maintain consistency across different rendering views and clearly differentiate between various objects and parts. These embeddings are transformed into a semantic map, which directs a region-specific SDS process, enabling precise optimization and compositional generation. By leveraging explicit semantic guidance, our method unlocks the compositional capabilities of existing pre-trained diffusion models, thereby achieving superior quality in 3D content generation, particularly for complex objects and scenes. Experimental results demonstrate that our SemanticSDS framework is highly effective for generating state-of-the-art complex 3D content.

## 1 Introduction

Generating high-quality 3D assets from textual descriptions is a long-standing goal in computer graphics and vision research. However, due to the scarcity of 3D data, existing text-to-3D generation models have primarily relied on leveraging powerful pre-trained 2D diffusion priors to optimize 3D representations, typically based on a score distillation sampling (SDS) loss (Poole et al., 2023). Notable examples include DreamFusion, which pioneered the use of SDS to optimize Neural Radiance Field (NeRF) representations (Mildenhall et al., 2021), and Magic3D (Lin et al., 2023a), which further advanced this approach by proposing a coarse-to-fine framework to enhance its performance.

Despite the advancements in lifting and SDS-based methods, generating complex 3D scenes with multiple objects or intricate interactions remains a significant challenge. Recent efforts have focused on incorporating additional guidance, such as box or layout information(Po & Wetzstein, 2024; Epstein et al., 2024; Zhou et al., 2024). Among them, Po & Wetzstein (2024) introduce locally conditioned diffusion for compositional scene diffusion based on input bounding boxes with one shared NeRF representation while Epstein et al. (2024) instantiate and render multiple NeRFs for a given scene using each NeRF to represent a separate 3D entity with a set of layouts. Further advancing this field, GALA3D (Zhou et al., 2024) utilizes large language models (LLMs) to generate coarse layouts to guide 3D generation for compositional scenes.

However, existing layout-guided compositional methods often fall short in achieving fine-grained control over the generated 3D scenes. The current form of box or layout guidance is relatively coarse and lacks the expressiveness required to effectively guide the SDS process in optimizing the intricate interactions or intersecting parts between multiple objects, particularly when generating objects with multiple attributes. This limitation stems from the fact that pre-trained 2D diffusion models, which are used in SDS, struggle to estimate accurate scores for complex scenarios with consistent views

GraphDreamer    LucidDreamer    GALA3D    GSGEN    **Ours**

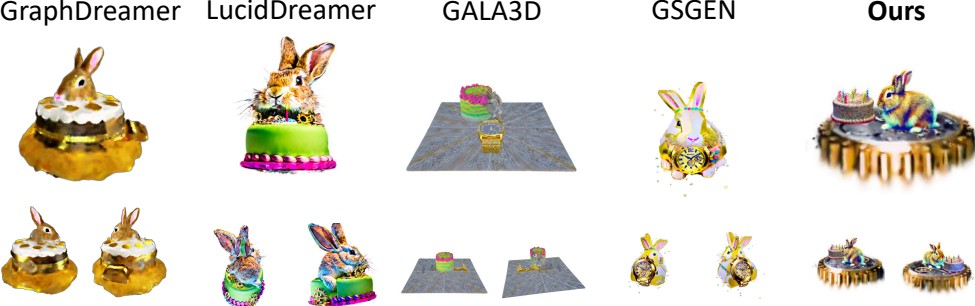

A rabbit sits atop a large, expensive watch with many shiny gears, made half of iron and half of gold, eating a birthday cake that is in front of the rabbit

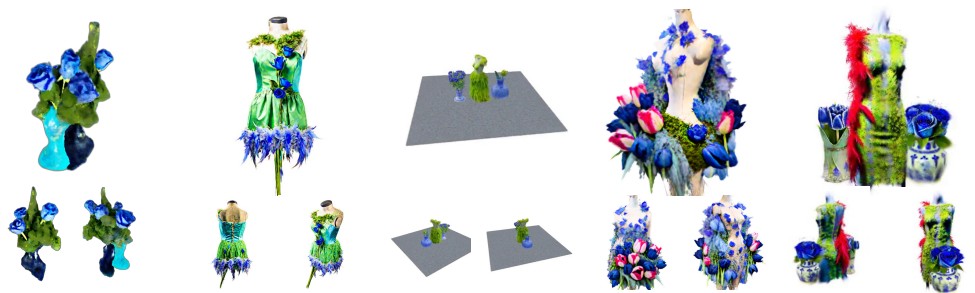

A mannequin adorned with a dress made of feathers and moss stands at the center, flanked by a vase with a single blue tulip and another with blue roses.

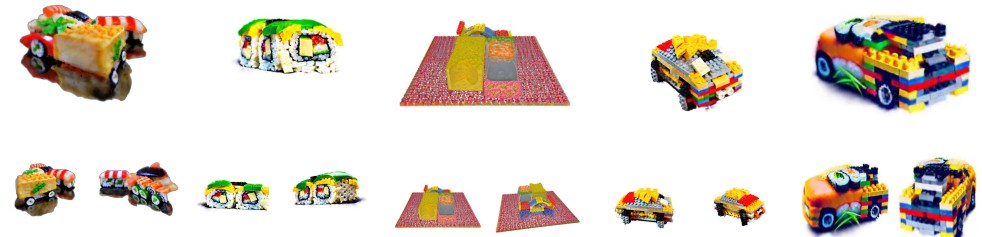

A car with the front right side made of cheese, the front left side made of sushi, and the back made of LEGO.

Figure 1: SEMANTICSDS achieves superior compositional text-to-3d generation results over state-of-the-art baselines, particularly in generating multiple objects with diverse attributes.

when explicit spatial guidance is absent (Li et al., 2023; Shi et al., 2024). As a result, the generated 3D scenes may lack the level of detail and realism desired, highlighting the need for more precise guidance mechanisms that can provide finer-grained control over the generation process.

To address these limitations, we propose Semantic Score Distillation Sampling (SEMANTICSDS), which boosts the expressiveness and precision of compositional text-to-3D generation. For more explicit 3D expression, we equip SEMANTICSDS with 3D Gaussian Splatting (3DGS) (Kerbl et al., 2023) as the 3D representation. Our approach consists of three key steps: (1) Given a text prompt, we propose a program-aided approach to improve the accuracy of LLM-based layout planning for 3D scenes. (2) We introduce novel semantic embeddings that remain consistent across various rendering views and explicitly distinguish different objects and parts. (3) We then render these semantic embeddings into a semantic map, which serves as guidance for a region-wise SDS process, facilitating fine-grained optimization and compositional generation. Our approach addresses the challenge of leveraging pre-trained diffusion models, which possess powerful compositional diffusion priors but are difficult to utilize (Wang et al., 2024a; Yang et al., 2024). By using explicit semantic map guidance, we innovatively unlock these compositional 2D diffusion priors for high-quality 3D content generation.

Our main contributions are summarized as follows:

- We propose SEMANTICSDS, a novel semantic-guided score distillation sampling approach that effectively enhances the expressiveness and precision of compositional text-to-3D generation, as shown in Figure 1.
- We introduce program-aided layout planning to improve positional and relational accuracy in generated 3D scenes, deriving precise 3D coordinates from ambiguous descriptions.
- We develop expressive semantic embeddings to augment 3D Gaussian representations, and propose a region-wise SDS process with the rendered semantic map, distinguishing different objects and parts in the compositional generation process.

## 2 RELATED WORK

**Text-to-3D Generation** Different approaches have been developed to achieve text-to-3D content generation (Deitke et al., 2024; Zeng et al., 2023), such as employing multi-view diffusion models (Shi et al., 2024; Wu et al., 2024a; Kong et al., 2024; Blattmann et al., 2023), direct 3D diffusion models (Gupta et al., 2023; Shue et al., 2023; Wu et al., 2024b) and large reconstruction models (Hong et al., 2024). For instance, multi-view diffusion models are trained and optimized by fine-tuning video diffusion on 3D datasets, aiding in 3D reconstruction (Voleti et al., 2024; Chen et al., 2024d; Han et al., 2024b). You et al. (2024) propose a training-free method that employs video diffusion as a zero-shot novel view synthesizer. However, these methods require numerous 3D data for training. In contrast, Score Distillation Sampling (SDS) (Poole et al., 2023; Wang et al., 2023) is 3D data-free and generally produces higher quality assets. SDS approaches harness the creative potential of 2D diffusion and have achieved significant advancements (Wang et al., 2024b; Yang et al., 2023b; Hertz et al., 2023), resulting in realistic 3D content generation and enhanced resolution of generative models (Zhu et al., 2024). In this paper, we propose a new SDS paradigm, namely SEMANTICSDS, for text-to-3D generation in complex scenarios, which first incorporates explicit semantic guidance into the SDS process.

**Compositional 3D Generation** Modeling compositional 3D data distribution is a fundamental and critical task for generative models. Current feed-forward methods (Shue et al., 2023; Shi et al., 2024) are primarily capable of generating single objects and face challenges when creating more complex scenes containing multiple objects due to limited training data. Po & Wetzstein (2024) fix the layout in multiple 3D bounding boxes and generate compositional assets with bounding-box-specific SDS. Recently, a series of learnable-layout compositional methods have been proposed (Epstein et al., 2024; Vilesov et al., 2023; Han et al., 2024a; Chen et al., 2024b; Li et al., 2024; Yan et al., 2024; Gao et al., 2024) . These methods combine multiple object-ad-hoc radiance fields and then optimize the positions of the radiance fields from external feedback. For example, Epstein et al. (2024) propose learning a distribution of reasonable layouts based solely on the knowledge from a large pre-trained text-to-image model. Vilesov et al. (2023) introduce an optimization method based on Monte-Carlo sampling and physical constraints. Non-learnable layout methods like (Zhou et al., 2024) and Lin et al. (2023b) further utilize LLMs or MLLMs to convert text into reasonable layouts. However, the current form of layout guidance is relatively coarse and not expressive enough for fine-grained control. We address this problem by incorporating semantic embeddings that ensure view consistency and distinctly differentiate objects into SDS processes, which are flexible and expressive for optimizing 3D scenes.

## 3 PRELIMINARIES

**Compositional 3D Gaussian Splatting** 3D Gaussian Splatting explicitly represents a 3D scene as a collection of anisotropic 3D Gaussians, each characterized by a mean $\mu \in \mathbb{R}^3$ and a covariance matrix $\Sigma$ (Kerbl et al., 2023). The Gaussian function $G(x)$ is defined as:

$$G(x) = \exp\left(-\frac{1}{2}(x-\mu)^\top \Sigma^{-1}(x-\mu)\right) \tag{1}$$

Rendering a compositional scene necessitates a transformation from object to composition coordinates, involving a rotation $\mathbf{R} \in \mathbb{R}^{3\times3}$, translation $\mathbf{t} \in \mathbb{R}^3$, and scale $s \in \mathbb{R}$ (Zhou et al., 2024;

Vilesov et al., 2023). This transformation is applied to the mean and variance of individual Gaussians, transitioning from the object's local coordinates to global coordinates: $\mu^{\text{global}} = s\mathbf{R}\mu^{\text{local}} + \mathbf{t}$, $\mathbf{\Sigma}^{\text{global}} = s^2\mathbf{R}\mathbf{\Sigma}^{\text{local}}\mathbf{R}^{\top}$.

For optimized rendering of compositional 3D Gaussians into 2D image planes, a tile-based rasterizer enhances rendering efficiency. The rendered color at pixel $v$ is computed as follows:

$$\mathbf{I}(v) = \sum_{i \in \mathcal{N}} c_i \alpha_i \prod_{j=1}^{i-1}(1 - \alpha_j), \tag{2}$$

where $c_i$ represents the color of the $i$-th Gaussian, $\mathcal{N}$ denotes the set of Gaussians within the tile, and $\alpha_i$ is the opacity.

**Score Distillation Sampling**   Yang et al. (2023a); Wang et al. (2023) have introduced a method to leverage a pretrained diffusion model, $\epsilon_\phi(x_t; y, t)$, to optimize the 3D representation, where $x_t$, $y$, and $t$ signify the noisy image, text embedding, and timestep, respectively.

Let $g$ represent the differentiable rendering fcuntion, $\theta$ denote the parameters of the optimizable 3D representation and $\mathbf{I} = g(\theta)$ be the resulting rendered image. The gradient for optimization is performed via Score Distillation Sampling:

$$\nabla_\theta \mathcal{L}_{\text{SDS}} = \mathbb{E}_{\epsilon,t}\left[ w(t)\left(\epsilon_\phi\left(x_t; y, t\right) - \epsilon\right) \frac{\partial \mathbf{I}}{\partial \theta} \right] \tag{3}$$

where $\epsilon$ is Gaussian noise and $w(t)$ is a weighting function. In compositional 3D generation, local object optimizations and global scene optimizations alternate in a compositional optimization scheme (Zhou et al., 2024). During local optimization, the parameters $\theta$ include the mean, covariance, and color of individual Gaussians. In global scene optimization, the parameters $\theta$ additionally include transformations—translation, scale, and rotation—that convert local to global coordinates.

## 4 METHOD

### 4.1 PROGRAM-AIDED LAYOUT PLANNING

A detailed characterization of multiple objects' positions, dimensions, and orientations requires numerous parameters, especially when additionally describing distinct attributes of various object components. In scenarios involving multiple objects, utilizing Large Language Models (LLMs) to derive precise 3D coordinates from ambiguous descriptions within a scene is often challenging. This difficulty arises because purely 3D numerical data and corresponding natural language descriptions do not frequently co-occur in the training data of LLMs (Hong et al., 2023; Xu et al., 2023). Consequently, issues such as overlapping objects or excessive distances between them may occur, particularly during interactions among objects. Therefore, we propose to leverage programs as the intermediate reasoning and planning steps (Gao et al., 2023) to effectively mitigate these challenges.

Let $y_c$ represent the complex user input, which includes multiple objects with various attributes. First, We utilize Large Language Models to identify all objects $\{O_k\}_{k=1}^{K}$ within $y_c$, where $K$ denotes the total number of objects. For each object, the corresponding prompt $y_k$ is recognized, and its dimensions are estimated. This includes considering the object's real-world size and its relationship with other objects to determine its relative size, facilitating the placement of all objects within the same scene.

Subsequently, LLMs sequentially position each object within the scene. In designing each object's placement, LLMs articulate the spatial relationships with relevant entities using programmable language descriptions that explicitly outline all mathematical calculations. This language is then converted into a program executed by a runtime, such as a Python interpreter, to produce the layout solution. These layouts, which include scale factors, Euler angles, and translation vectors, are employed to transform 3D Gaussians from local coordinates to global coordinates during rendering.

Furthermore, for each object $O_k$, LLMs decompose its layout space into $n_k$ complementary regions, each with distinct attributes and different subprompts $\{y_{k,l}\}_{l=1}^{n_k}$. These complementary regions are

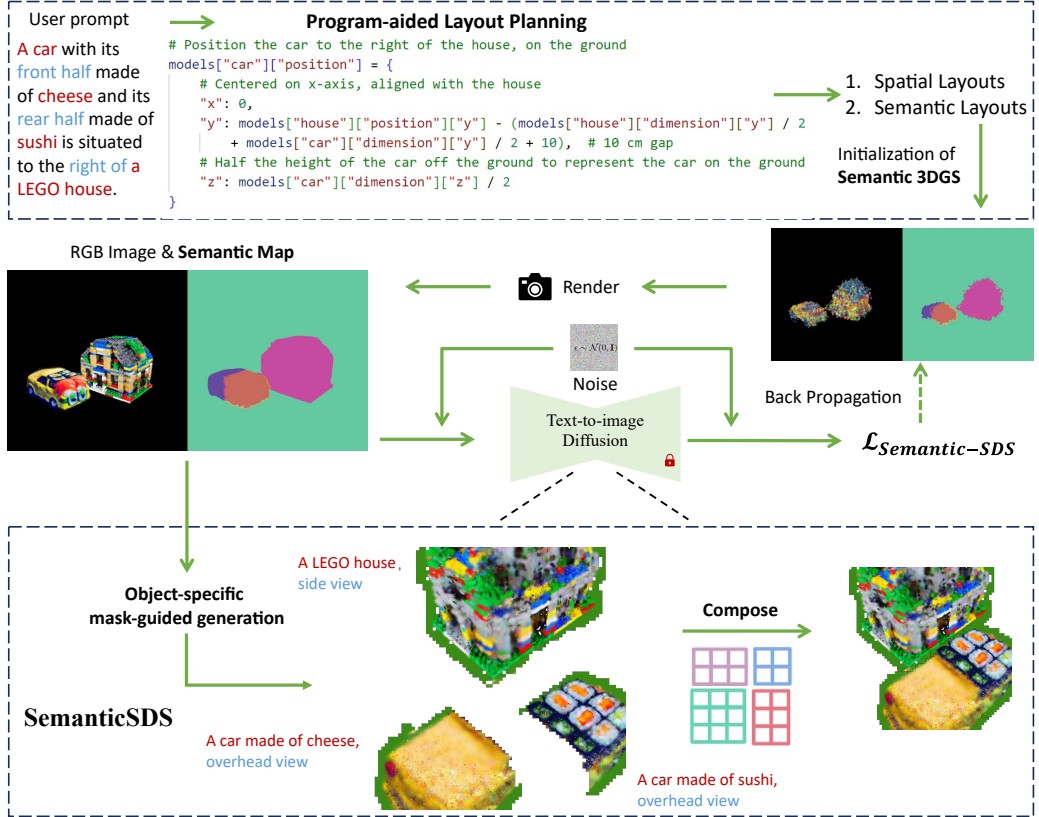

Figure 2: Overview of SEMANTICSDS, comprising of program-aided layout planning (top) and regional denoising with semantic map (bottom).

designed to be non-overlapping and collectively encompass the entire layout space of their respective object. To generate meaningful and accurate complementary regions, LLMs employ a structured decomposition process that segments the space of object $O_k$ into hierarchical divisions based on depth, width, and length dimensions. This process is documented using programmable language descriptions and subsequently converted by the program into 3D bounding boxes within a normalized coordinate system, where the coordinates range between 0 and 1. Details on the prompts used for this program-aided layout planning are provided in Appendix A.1.

## 4.2 SEMANTIC SCORE DISTILLATION SAMPLING

**Prompt-Guided Semantic 3D Gaussian Representation** To generate 3D scenes involving multiple objects with diverse attributes and to precisely control the attributes of distinct spatial regions within each object, it is essential to utilize features that represent the fine-grained semantics of 3D Gaussians. We design new prompt-guided semantic 3D Gaussian representations. During initialization, the subprompt $y_{k,l}$ corresponding to the $i$-th Gaussian is encoded via the CLIP text encoder $\Phi$ (Radford et al., 2021) to obtain the high-dimensional semantic embedding, $\mathbf{h}_i = \Phi(y_{k,l}) \in \mathbb{R}^{d_\mathbf{h}}$. Given the significant memory demands imposed by the large dimensions of $d_\mathbf{h}$, a lightweight autoencoder is employed. This autoencoder effectively compresses the scene's high-dimensional semantic embeddings into more manageable, low-dimensional representations, represented as $\mathbf{f}_i = E(\mathbf{h}_i) \in \mathbb{R}^{d_\mathbf{f}}$. The loss function for the autoencoder is defined as:

$$\mathcal{L}_{ae} = \sum_{i \in \mathcal{N}} d_{ae}(D(E(\mathbf{h}_i)), \mathbf{h}_i) \tag{4}$$

where $d_{ae}$ denotes the metric combining the $\mathcal{L}_1$ loss and the symmetric cross entropy loss from CLIP (Radford et al., 2021).

For each object $O_k$, we utilize Shap-E (Jun & Nichol, 2023) to generate the positions of Gaussians based on the corresponding prompt $y_k$. Recall that the program-aided layout planning decomposes

the layout for object $O_k$ into $n_k$ complementary 3D bounding boxes within a normalized coordinate system, each associated with different subprompts $\{y_{k,l}\}_{l=1}^{n_k}$. After transforming the Gaussians into the normalized coordinate system, we enhance the Gaussians within the 3D bounding boxes corresponding to subprompt $y_{k,l}$ with semantic embeddings $E(\Phi(y_{k,l}))$. Subsequently, we transform the Gaussians to global coordinates using the scale factors, Euler angles, and translation vectors specified in the layout for object $O_k$.

Represent the semantic embedding of the $i$-th Gaussian as $\mathbf{f}_i \in \mathbb{R}^d$. The semantic information is then integrated into the rendered 2D image by rendering the semantic embedding at pixel $v$ using the formula:

$$\mathbf{F}(v) = \sum_{i \in \mathcal{N}} \mathbf{f}_i \alpha_i \prod_{j=1}^{i-1} (1 - \alpha_j) \tag{5}$$

The rendered semantic embedding $\mathbf{F}(v)$, derived from equation 5, is fed into the decoder $D$ to reconstruct $\mathbf{S}(v) = D(\mathbf{F}(v)) \in \mathbb{R}^{d_{\mathbf{h}}}$ and then generates a semantic map $\mathbf{S} \in \mathbb{R}^{H \times W \times d_{\mathbf{h}}}$ indicating the rendered image's semantic attributes.

**Semantic Score Distillation Sampling**   To enable fine-grained controllable generation, the generated semantic map is integrated into the spatial composition of scores for distillation sampling. The subprompt $y_{k,l}$ is processed through the CLIP text encoder $\Phi$ to produce the subprompt embedding $\mathbf{q}_{k,l} = \Phi(y_{k,l}) \in \mathbb{R}^{d_{\mathbf{h}}}$. The probability that pixel $v$ corresponds to subprompt $y_{k,l}$ is computed as:

$$p(k, l \mid v) = \frac{\exp\left(\cos\left(\mathbf{q}_{k,l}, \mathbf{S}(v)\right) / \tau\right)}{\sum_{k'=1}^{K} \sum_{l'=1}^{n_{k'}} \exp\left(\cos\left(\mathbf{q}_{k,l}, \mathbf{S}(v)\right) / \tau\right)} \tag{6}$$

where $\tau$ is a temperature parameter learned by CLIP and $\cos(\cdot, \cdot)$ denotes cosine similarity. This facilitates the derivation of the mask $\mathbf{M}_{k,l}(v)$, which indicates whether the semantic properties of pixel $v$ align with subprompt $y_{k,l}$.

$$\mathbf{M}_{k,l}(v) = \begin{cases} 1 & \text{if } (k, l) = \arg\max_{k',l'} p\left(k', l' \mid v\right) \\ 0 & \text{otherwise} \end{cases} \tag{7}$$

The semantic mask $\mathbf{M}_{k,l} \in \{0, 1\}^{H \times W}$ is subsequently utilized to guide the score distillation sampling. To ensure that the Gaussians near the edges of objects are not overlooked, the mask $\mathbf{M}_{k,l}$ is subjected to a max pooling operation with a $5 \times 5$ kernel, resulting in $\hat{\mathbf{M}}_{k,l}$. Although diffusion models generally lack an inherent distinction at the object and part levels in their latent spaces or attention maps for fine-grained control (Lian et al., 2024), recent advancements in compositional 2D image generation have implemented spatially-conditioned generation (Chen et al., 2024a; Yang et al., 2024; Xie et al., 2023). This is achieved through regional denoising or attention manipulation, allowing for fine-grained control over the semantics of the generated images. Specifically, the overall denoising score is calculated as the aggregate of the individually masked denoising scores for each visible subprompt $y_{k,l}$:

$$\hat{\epsilon}_\phi\left(x_t; \mathbf{y}, t\right) = \mathbb{E}_{k,l}\left[\epsilon_\phi\left(x_t; y_{k,l}, t\right) \odot \hat{\mathbf{M}}_{k,l}\right] \tag{8}$$

where $\odot$ denotes element-wise multiplication. Instead of conditioning the diffusion models on a single text prompt, our semantic score distillation sampling employs the compositional denoising score as follows:

$$\nabla_\theta \mathcal{L}_{\text{SemanticSDS}} = \mathbb{E}_{\epsilon,t}\left[w(t)\left(\hat{\epsilon}_\phi\left(x_t; \mathbf{y}, t\right) - \epsilon\right) \frac{\partial \mathbf{x}}{\partial \theta}\right] \tag{9}$$

In this way, we can sufficiently leverage expressive compositional generation prior of pretrained 2D diffusion models for text-to-3D generation. More details on SEMANTICSDS are provided in Appendix A.2.

**Object-Specific View Descriptor for Global Scene Optimization**   Unlike object-centric optimization, scenes do not exhibit distinct perspectives as individual objects do. Effective scene generation necessitates precise, part-level control over the optimization of distinct object views. Terms such as "side view" or "back view" are rarely applicable to multi-object scenes, and pretrained diffusion models often struggle to generate images accurately from such prompts (Li et al., 2023).

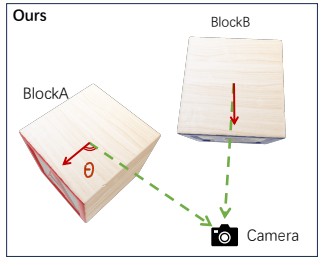 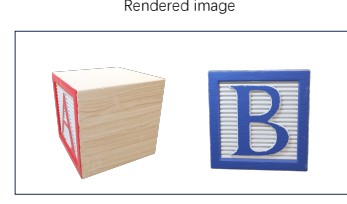

Prompt: A photo of blockA and blockB, front view.
Prompt 1: A photo of blockA, **side view**.
Prompt 2: A photo of blockB, **front view**.

Figure 3: Illustration of our proposed object-specific view descriptor for global scene optimization.

Moreover, within a single rendered image, different objects may be visible from varying perspectives. Using a unified view descriptor for an entire scene with multiple objects exacerbates the Janus Problem (Poole et al., 2023). Although the compositional optimization scheme alternates between local object optimizations and global scene optimizations (Zhou et al., 2024), allowing for the correct optimization of different views of objects in local coordinates, it is confounded by optimizations under global coordinates. This limits the frequency of global scene optimizations and results in a lack of scene coherence, harmony, and lighting consistency.

To address this issue, in our SEMANTICSDS, we append an object-specific view descriptor $y_k^{\text{view}}$ to the corresponding subprompts $\{y_{k,l}\}_{l=1}^{n_K}$ to optimize individual objects within the rendered image (in Figure 3). The same view descriptor $y_k^{\text{view}}$ is consistently applied across different parts of each multi-attribute object. Specifically, we determine the camera's elevation and azimuth angles relative to each object by computing the angle between the vector $\hat{n}$, which extends from the object to the camera, and specific reference axis vectors, such as the positive z-axis. This calculation facilitates the selection of the most appropriate object-specific view descriptor. For instance, if the angle between $\hat{n}$ and the positive $z$-axis remains below a predefined threshold, indicative of a high azimuth angle, the descriptor $y_k^{\text{view}}$ is assigned as an overhead view descriptor for that object.

## 5 EXPERIMENTS

**Implementation Details.** The guidance model is implemented using the publicly accessible diffusion model, StableDiffusion (Rombach et al., 2022), specifically utilizing the checkpoint *runwayml/stable-diffusion-v1-5*. Positions of the Gaussians are initialized using Shap-E (Jun & Nichol, 2023), with each object initially comprising 12288 Gaussians. Optimization processes are performed against randomly selected, uniformly colored backgrounds. For densification, Gaussian components are split based on the gradient of their position in view space, using a threshold $T_{\text{pos}} = 2$. Compactness-based densification is then applied every 2000 iterations, involving each Gaussian and one of its nearest neighbors, as described by GSGEN (Chen et al., 2024c). Pruning involves the removal of Gaussians exhibiting an opacity lower than $\alpha_{\min} = 0.3$, as well as those with excessively large radii in either world-space or view-space, at intervals of every 200 iterations.

Training alternates between local and global optimization. During global optimization, the rendered objects vary by switching between the entire scene and pairs of objects. Camera sampling maintains the same focal length, elevation, and azimuth range as specified in (Chen et al., 2024c). The threshold for the selection of object-specific view descriptors includes: an overhead view descriptor chosen for elevation angles exceeding $60°$, a front view descriptor selected for azimuth angles within $±45°$ of the positive x-axis, and a back view descriptor utilized for $±45°$ angles on the negative x-axis.

Table 1: Quantitative Comparison

| Metrics | GraphDreamer | GSGEN | LucidDreamer | GALA3D | SemanticSDS (Ours) |
|---|---|---|---|---|---|
| CLIP Score ↑ | 0.289 | 0.314 | 0.311 | 0.305 | **0.321** |
| Prompt Alignment ↑ | 56.9 | 63.3 | 64.4 | 85.0 | **91.1** |
| Spatial Arrangement ↑ | 53.8 | 62.8 | 65 | 80.0 | **85.7** |
| Geometric Fidelity ↑ | 53.8 | 71.1 | 71.8 | 80.3 | **83.0** |
| Scene Quality ↑ | 54.9 | 71.2 | 65.9 | 82.3 | **86.9** |

GraphDreamer  LucidDreamer  GALA3D  GSGEN  **Ours**

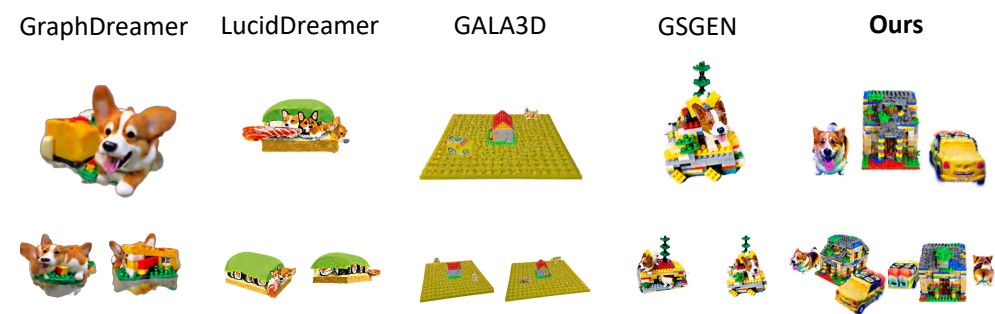

A corgi is positioned to the left of a LEGO house, while a car with its front half made of cheese and its rear half made of sushi is situated to the right of the house made of LEGO.

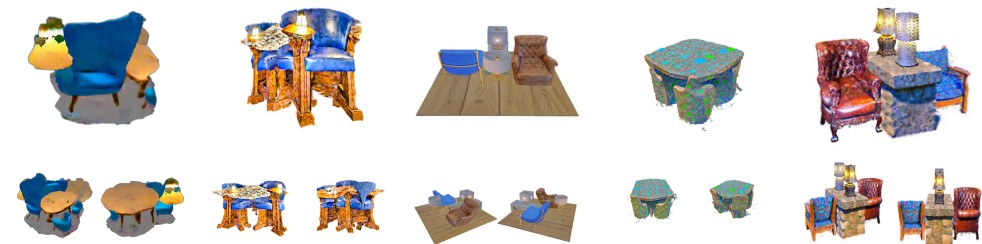

In a library's reading room, a stone block table is flanked by two types of chairs: a high-back leather chair on the left side and a low-slung, blue chair on the right. Two lamps, one with a classic design and the other with a modern aesthetic, are positioned above the table to provide lighting.

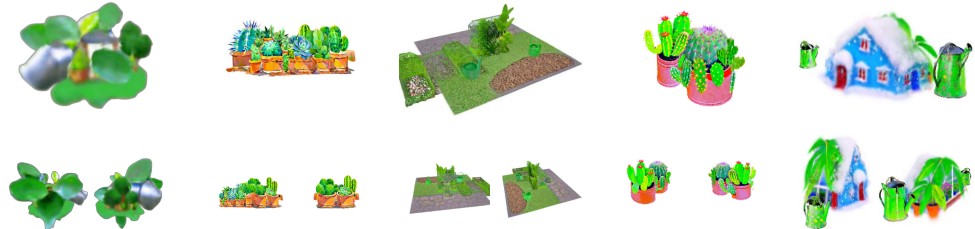

In a botanic garden, a greenhouse is split into two climates. The left side is a tropical environment with lush greenery, and the right side is an icy snowy climate with cacti and succulents. Two watering cans, one large and the other small, are placed at the entrance.

Figure 4: **Qualitative comparisons of text-to-3D generation.** Comparison results demonstrate that SEMANTICSDS synthesizes more precise and realistic multi-object scenes with better visual details, geometric expressiveness, and semantic consistency.

**Baseline methods.** To evaluate the performance of SEMANTICSDS on the complex Text-to-3D task involving multiple objects with varied attributes, we compare it with state-of-the-art (SOTA) methods. These include the compositional 3D generation method GALA3D (Zhou et al., 2024) and GraphDreamer (Gao et al., 2024), noted for their ability to generate intricate scenes with multiple objects. Additionally, we consider GSGEN (Chen et al., 2024c) and LucidDreamer (Liang et al., 2024), both are capable of producing high-quality, complex objects with diverse attributes.

**Metrics.** CLIP Score (Radford et al., 2021) is employed as the evaluation metric to assess the quality and consistency of the generated 3D scenes with textual descriptions. However, CLIP tends to focus on the primary objects within the rendered image, and when used to evaluate complex text-to-3D tasks involving multiple objects with varied attributes, it may not adequately assess the geometry of all objects or the rationality of their spatial arrangements. This limitation results in a

misalignment with human judgment regarding evaluation criteria. Therefore, following Wu et al. (2024c), GPT-4V is utilized as a human-aligned evaluator to compare 3D assets based on predefined criteria. These criteria include: (1) Prompt Alignment: ensuring that all objects specified in the user prompts are present and correctly quantified; (2) Spatial Arrangement: evaluating the logical and thematic spatial arrangement of objects; (3) Geometric Fidelity: assessing the geometric fidelity of each object for realistic representation; and (4) Scene Quality: determining the overall scene quality in terms of coherence and visual harmony. More details on metrics are provided in the Appendix A.3.

## 5.1 MAIN RESULTS

**Quantitative Analysis**    To evaluate the performance of SEMANTICSDS in Text-to-3D tasks involving multiple objects with varied attributes, quantitative metrics were employed. The scene prompts used for evaluation are shown in Table 2 and Table 3. As shown in Table 1, the CLIP Score indicates that SEMANTICSDS exhibits strong alignment with the primary semantics of user prompts. Specifically, SEMANTICSDS excels in Prompt Alignment, ensuring that all objects specified in user prompts are present and correctly quantified. Additionally, it demonstrates superior performance in Spatial Arrangement, effectively designing the layout of interactive objects to support the scene's intended theme. Furthermore, by explicitly guiding SDS with rendered semantic maps, SEMANTICSDS achieves outstanding generation of individual objects with diverse attributes across different spatial components, resulting in high scores in object-level Geometric Fidelity. Additionally, the use of compositional 3D Gaussian Splatting for scene representation helps SEMANTICSDS to effectively disentangle objects within the scene. This, combined with explicit semantic guidance to the SDS, contributes to achieving the highest score in Scene Quality.

**Qualitative Analysis**    To intuitively demonstrate the superiority of the proposed method in generating complex 3D scenes with multiple objects possessing diverse attributes, a qualitative comparison with baseline models is conducted. As illustrated in Figure 4, GALA3D, with a compositional optimization scheme, successfully generates individual objects that align with user prompts. However, it fails to produce plausible results when objects have multiple attributes. Although GSGEN and LucidDreamer generate high-quality individual objects, the presence of multiple objects often leads to entanglement, compromising consistency with user prompts. Additionally, these models are unable to generate reasonable objects when individual objects possess numerous attributes. In contrast, SEMANTICSDS employs guided diffusion models with explicit semantics, effectively generating scenes that include multiple objects with diverse attributes. Moreover, by utilizing program-aided layout planning, SEMANTICSDS produces more coherent layouts than GALA3D in scenarios involving complex spatial relationships among multiple objects. For example, in Figure 1, both table lamps are correctly placed on the table without appearing to float when using SEMANTICSDS.

**User Study**    We conducted a user study to compare our method with baseline methods across 30 scenes involving about 160 objects. The scene prompts used are shown in Table 2 and Table 3. Each participant was shown a user prompt alongside 3D scenes generated by all methods simultaneously and asked to select the most realistic assets based on geometry, prompt alignment, and accurate placement. Figure 5 illustrates that SEMANTICSDS significantly outperformed previous methods in terms of human preference.

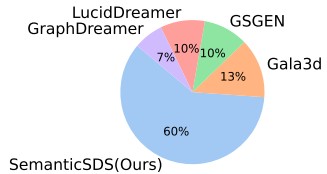

Figure 5: **User study results.** SEMANTICSDS is preferred 60% of the time by users than baseline methods.

## 5.2 MODEL ANALYSIS

**Effectiveness of Program-aided Layout Planning**    We assess the necessity of program-aided layout planning through an ablation study. The qualitative comparison of generated layouts is illustrated in Figure 6. Without program-aided planning, layout placement often lacks rationale and results in poor spatial arrangements. In contrast, the program-aided strategy positions the layouts logically and divides the layout into meaningful and precise complementary regions for objects with multiple attributes, resulting in an effective spatial arrangement.

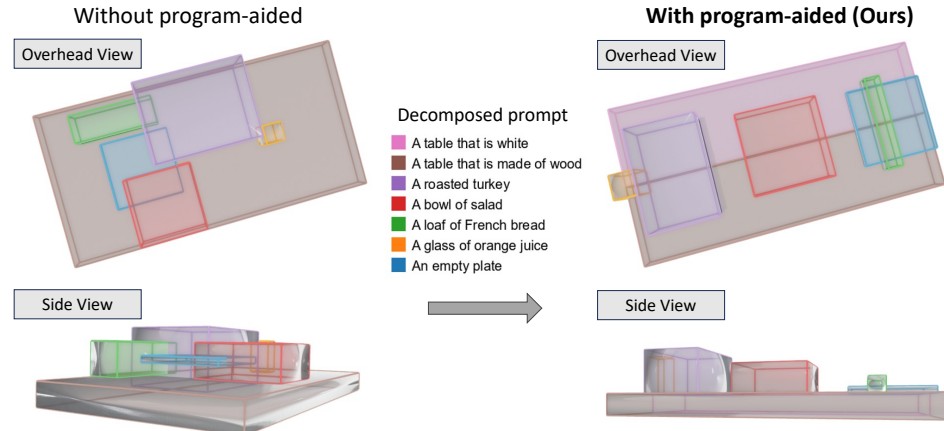

**Text prompt:** A table, half made of wood and half white, holds a roasted turkey, a salad, a glass of orange juice, and a plate with a loaf of French bread.

Figure 6: Qualitative comparisons between without and with our program-aided layout planning.

**Impact of Semantic Score Distillation Sampling** Ablation experiments are performed on Semantic Score Distillation Sampling to evaluate the effects of explicitly guiding SDS with rendered semantic maps. In Figure 7, without SEMANTICSDS, while objects with single attributes are generated effectively, those with varied attributes often experience blending issues. For instance, the "house" shows snow bricks mixed with LEGO bricks, failing to meet the user prompt's spatial requirements. The snow bricks are inaccurately represented as white LEGO bricks, which do not align with the intended attributes. Additionally, one attribute may dominate, causing others to disappear, such as in the "car" with three attributes in Figure 7. Conversely, SemanticSDS enables precise control over the attributes in distinct spatial regions of each object, producing objects with diverse attributes and smooth transitions between regions with different attributes.

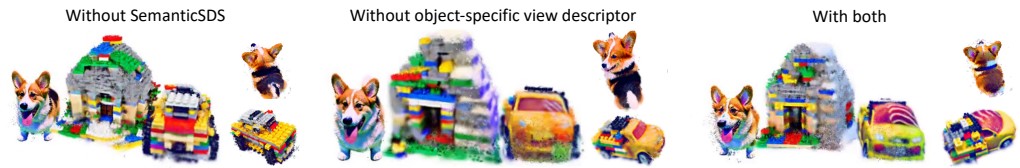

**Text prompt:** A corgi is positioned to the left of a house that is half made of LEGO and half of snow. To the right of the house, there is a car with its front right side made of cheese, front left side made of sushi, and the back made of LEGO.

Figure 7: Qualitative analysis. Our SEMANTICSDS provides more precise and fine-grained control and our proposed object-specific view descriptor helps with better multi-view understanding.

**Object-Specific View Descriptor** To assess the effectiveness of the object-specific view descriptor, we replace it with the scene-centric view descriptor utilized by GSGEN during global optimization. This change increases the occurrence of the Janus Problem, as illustrated by the overhead view of the corgi in the middle of Figure 7. These findings highlight the crucial role of selecting an appropriate view descriptor to enhance the plausibility of generated 3D scenes.

# 6 CONCLUSION

In this paper, we introduce SEMANTICSDS, a novel SDS method that significantly enhances the expressiveness and precision of compositional text-to-3D generation. By leveraging program-aided layout planning, semantic embeddings, and explicit semantic guidance, we unlock the compositional priors of pre-trained diffusion models and achieve realistic high-quality generation in complex scenarios. Our extensive experiments demonstrate that SEMANTICSDS achieves state-of-the-art results for generating complex 3D content. As we look to the future, we envision SEMANTICSDS as a foundation for even more applications, such as automatic editing and closed-loop refinement, paving the way for unprecedented levels of creativity and innovation in 3D content generation.

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

# A MORE IMPLEMENTATION DETAILS

## A.1 PROMPTS FOR PROGRAM-AIDED LAYOUT PLANNING

You are a proficient 3D scene designer with the ability to effectively position 3D models within a 3D cubic space. Using a provided scene description, please carry out the following tasks:

**1. Identify 3D Models:**
  - Identify and list the 3D models mentioned in the description.

**2. Write Python Code to Estimate Dimensions and Arrange 3D models:**
  - Measure the dimensions of each 3D model as if they were toys. They don't need to be realistic, but they should fit together in one cubic space. Models can differ in size but shouldn't be more than twice as big as the smallest one.
  - Arrange the identified 3D models in a 3D cubic space centered at coordinates [0, 0, 0], with measurements in centimeters. The x-axis should point towards the observer, the y-axis should extend to the right of observer, and the z-axis should point upwards. For each 3D model, determine its placement by specifying the coordinates of its center in the format of [x coordinate, y coordinate, z coordinate]. Ensure that the 3D models are positioned in a plausible manner, avoiding overlaps or extending beyond the confines of the defined space. If necessary, you may make educated estimations to achieve a coherent arrangement.

Here are some examples, follow the example to design the 3D scene:
Scene description: ......
Let's think step by step and write the python codes.

**Task 1: Identify Models.** Identify and list the 3D models mentioned in the description. If two models are closely associated, I will identify them as one model.
From the scene description, the tangible models mentioned are:
.....
```
models = {
    "corgi": {"object description": "Corgi dog"},
    "beret": {"object description": "Beret hat"},
    "house": {"object description": "The house straddles a divide that separates spring and winter horizontally."},
    "car": {"object description": "Car, with front layer made of wood and rear layer made of sushi and cheese. The left half of the rear layer is made of sushi, and the right half is made of cheese."}
}
```
**Task 2: Estimate Dimensions.** For the scene description that involves a corgi, a beret, a house, and a car with layered materials, let's estimate the dimensions of each model to ensure they fit within a unified cubic space.
```
models["corgi"]["dimension"] = {"x": 30, "y": 15, "z": 20}  # cm
models["beret"]["dimension"] = {"x": 15, "y": 15, "z": 5}  # cm, diameter fitting the corgi's head, thickness/height when laid flat
models["house"]["dimension"] = {"x": 45, "y": 30, "z": 35}  # cm, the size of a house can vary widely. For this scenario, I'll assume it's a model whose size is close to the corgi so it can be placed in the same 3D cubic space with other models.
models["car"]["dimension"] = {"x": 40, "y": 20, "z": 15}  # cm, toy car size fitting the scene
```
**Task3: Calculate the positions considering viewing from the front.** Arrange the identified 3D models in a 3D cubic space centered at coordinates [0, 0, 0]. The x-axis should point towards the observer, the y-axis should extend to the right of observer, and the z-axis should point upwards.
We'll start by placing the house at the center, then position the corgi and car relative to the house, and finally, place the beret on top of the corgi.

......

```
# Position the corgi to the left of the house, on the ground
models["corgi"]["position"] = {
    # Centered on x-axis, aligned with the house
    "x": 0,
    "y": models["house"]["position"]["y"] - (models["house"]["dimension"]["y"] / 2 + models["corgi"]["dimension"]["y"] / 2 + 5),
    # Half the height of the corgi off the ground to represent the corgi sitting on the ground
    "z": models["corgi"]["dimension"]["z"] / 2
}
......
```

Scene description: {{user_prompt}}
Let's think step by step and write the python codes.

Figure 8: The prompt for scene-level decomposition in program-aided layout planning.

Large Language Models (LLMs) have the potential for spatial awareness; however, precise 3D layout generation from vague language descriptions is challenging. This difficulty arises because 3D digital data and corresponding natural language descriptions often do not appear simultaneously (Hong et al., 2023; Xu et al., 2023). Moreover, minor numerical changes, which might not be reflected in imprecise language, can lead to unrealistic spatial arrangements of 3D scenes. Additionally, the spatial arrangement of multi-object scenes requires numerous parameters, making a program-aided approach necessary to bridge the gap between natural language descriptions and 3D digital data.

Specifically, we decompose the process of generating multiple objects with diverse attributes into two steps: scene-level decomposition and object-level decomposition. In scene decomposition, we guide LLMs to translate user prompts into Python programs, using explicit mathematical operations

As a 3D model designer, you are tasked with designing an object described in the user prompt. This object has multiple attributes, with different parts possessing different attributes. Your job is to divide the object as described in the user prompt into parts, each with a single attribute, and rewrite the corresponding prompt for each part. Specifically, you need to divide the 3D bounding box encompassing the object into different complementary smaller bounding boxes, and output in the specific format.
**# The specific format description**
The output should be a JSON object that represents the 3D bounding box of the object. This object should have a key named "depth split" that contains an array of objects. Each object represents a division of the bounding box along the depth axis. The object should have two keys: "size" and "vertical split". The "size" key represents the size of this part relative to other parts in the same split.
The "vertical split" key should contain an array of objects. Each object represents a division of the bounding box along the vertical axis. The object should have two keys: "size" and "horizontal split".
The "size" key represents the size of this part relative to other parts in the same split.
The "horizontal split" key should contain an array of objects. Each object represents a division of the bounding box along the horizontal axis. The object should have two keys: "size" and "prompt".
The "size" key represents the size of this part relative to other parts in the same split.
The "prompt" key should contain the prompt for the specific part of the object. The prompt should be a string that describes the part of the object and its single attributes.

# Examples
......

Figure 9: The prompt for decomposing each object into complementary regions.

to represent relationships between objects. For object decomposition, since complementary regions are designed to be non-overlapping and collectively encompass the entire layout space of their respective objects, we devised a scheme employing structured JavaScript Object Notation (JSON) to represent hierarchical divisions based on depth, width, and length dimensions. Figures 8 and 9 illustrate the detailed prompts for scene and object decomposition, respectively.

## A.2 SEMANTICSDS

**Camera Sampling** Training alternates between local and global optimization. During local optimization, objects are not transformed into global coordinates. In global optimization, the rendering of objects varies by switching between the entire scene and pairs of objects to better optimize those that interact or occlude each other. When rendering only a pair of objects, the camera's look-at point is sampled at the midpoint between the two objects rather than the center of the entire scene. Additionally, we apply a dynamic camera distance from the object pair to ensure the objects are appropriately sized in the rendered images. Specifically, the camera distance is determined by the scale of the objects and the distance between their centers.

**Pooling of Semantic Masks** Given that the rendered RGB images and the semantic map have sizes of $512 \times 512$, whereas the latents for denoising are of size $64 \times 64$, we convert the semantic map $\mathbf{S}$ into masks to compose the denoising scores predicted by diffusion models. Subsequently, for each mask $\mathbf{M}_{k,l} \in \{0,1\}^{512 \times 512}$, we apply average pooling with a stride of 8 using an $8 \times 8$ kernel to downsample the data. To ensure that Gaussians near the edges of objects and isolated Gaussians are not overlooked, the mask $\mathbf{M}_{k,l}$ undergoes a max pooling operation with a $5 \times 5$ kernel, resulting in $\hat{\mathbf{M}}_{k,l}$.

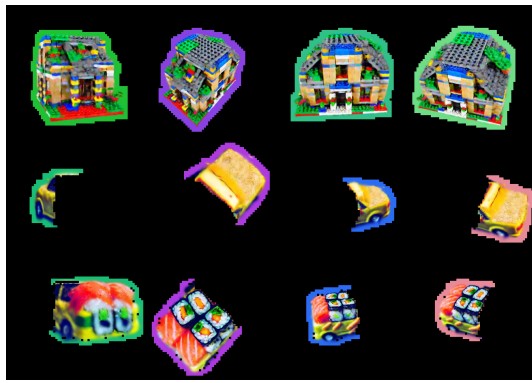

Figure 10: Visualization of semantic masks.

## A.3 DETAILS OF METRICS

Table 2: The scene prompts used for the quantitative analysis and user study. (Part 1)

| |
|---|
| A bedroom with a bed, two wooden nightstands, a square wooden table with a table lamp on it, and a wooden wardrobe. |
| Two toy teddy bears sit on a wooden treasure chest, one of them dressed in superhero attire. |
| A bookshelf divided into sections with one half featuring blue metal shelves and the other half solid oak panels displays an array of colorful novels, a wooden globe, and a terracotta planter with green ivy. |
| On the coffee table in front of the sofa, there is a ceramic teapot and two teacups. Next to the sofa, there is a floor lamp. |
| A corgi is positioned to the left of a LEGO house, while a car with its front half made of cheese and its rear half made of sushi is situated to the right of the house made of LEGO. |
| There are several pieces of cheese and some bunches of grapes next to a bottle of red wine and two wine glasses. |
| A table with a roasted turkey, a salad, a loaf of French bread, a glass of orange juice. |
| A castle made of snow bricks and stone bricks is next to a train with a front made of cake and a back made of a steam engine. |
| A car with the front right side made of cheese, the front left side made of sushi, and the back made of LEGO. |
| A rabbit sits atop a large, expensive watch with many shiny gears, made half of iron and half of gold, eating a birthday cake that is in front of the rabbit. |
| A vintage wooden chair with one armrest upholstered in navy blue fabric and the other in leather sits beside a stack of hardcover books and a metallic desk organizer holding various stationery items. |
| A cozy scene with a plush triceratops toy surrounded by a plate of chocolate chip cookies, a glistening cinnamon roll, and a flaky croissant. |
| At the head of the table sits a plush dragon toy and a plate of fried chicken and waffles. The centerpiece is a roasted turkey, flanked by sushi and pyramid-shaped tacos, creating a fantastical banquet scene. |
| In a library's reading room, a stone block table is flanked by two types of chairs: a high-back leather chair on the left side and a low-slung, blue chair on the right. Two lamps, one with a classic design and the other with a modern aesthetic, are positioned above the table to provide lighting. |
| There is a plate of fresh strawberries and a glass bottle of milk. To the left, there is a woven basket filled with eggs, and to the right, there is a ceramic teapot next to a small jar of honey. |
| At the center is a sliced loaf of bread, surrounded by pancakes covered in maple syrup, a croissant, and a glazed cinnamon roll. A vase of sunflowers adds a fairy-tale atmosphere. |
| A teapot and two teacups are placed together. To the right, there is a plate of cookies and a fruit bowl filled with fresh fruits. |
| On a round wooden table, there is a clay vase filled with blooming sunflowers. To the right of the table, there is a computer monitor. |
| Nearby an origami motorcycle, there is a complex watch mechanism and an intricately carved wooden knight chess piece, creating a scene that combines art and precision craftsmanship. |
| A hamburger, a loaf of bread, an order of fries, and a cup of Coke. |
| In a botanic garden, a greenhouse is split into two climates. The left side is a tropical environment with lush greenery, and the right side is an icy snowy climate with cacti and succulents. Two watering cans, one large and the other small, are placed at the entrance. |
| A wooden knight chess piece hovering above a dress made of pink feathers. |
| There is an antique typewriter with a brass desk lamp to its left and a stack of thick books to its right. |
| A sleek desk lamp with a lampshade made of woven bamboo and a base crafted from brushed stainless steel stands next to an open notebook and a ceramic mug filled with colorful pens. |

**CLIP Score** The CLIP score utilizes CLIP embeddings (Radford et al., 2021) to evaluate text-to-3D alignment. Following previous methods (Zhou et al., 2024; Gao et al., 2024), we calculate the

Table 3: The scene prompts used for the quantitative analysis and user study. (Part 2)

| |
|---|
| A mannequin adorned with a dress made of feathers and moss stands at the center, flanked by a vase with a single blue tulip and another with blue roses. |
| On a table, there is a compass and a flashlight. |
| In a community plaza, a pair of statues stands facing each other. One statue is a representation of a historical figure cast in bronze, and the other is a modern, abstract sculpture made of mirrored glass. |
| There is an iron pot with hot soup cooking inside. To the left of the campfire, there is a tree stump with a pair of hiking boots on top. |
| A puppy is lying on the iron plate at the top of the Great Pyramid, which is made of snow bricks and stone bricks. |
| A glass block, a wooden block, a stone block, and a glowing lamp are displayed. They are arranged sequentially from left to right: the wooden block is first, followed by the stone block, then the glass block, and the glowing lamp is placed at the back of the stone block. |

cosine similarity between the user prompt and scene images rendered from different perspectives. For each scene, we take the maximum CLIP score from all rendered images as the representative score. We then compare the average of these maximum scores across different scenes for each method.

**GPT-4V as A Human-Aligned Evaluator** Due to the limitations of the CLIP score in capturing spatial arrangement and geometric fidelity, we follow Wu et al. (2024c) and employ GPT-4V to evaluate complex 3D scenes involving multiple objects with varied attributes. Specifically, we provide GPT-4V with rendered images of the same 3D scene generated by different methods and require it to score each scene on four aspects: Prompt Alignment, Spatial Arrangement, Geometric Fidelity, and Scene Quality, each on a scale from 1 to 100. For each scene and method pair, we perform three independent evaluations. The final score for each method is obtained by averaging the scores across different scenes and comparisons with other methods. Figure 11 presents the prompt used to guide the GPT-4V evaluator. In the prompt, "method A" and "method B" are used to anonymize the methods, preventing name bias in GPT-4V's judgment.

## B    MORE SYNTHESIS RESULTS

We present further comprehensive results of SEMANTICSDS for generating intricate and complex scenes involving multiple objects, as shown in Figures 14 and 16. Figure 15 also demonstrates SEMANTICSDS's fine control over detailed objects with varied attributes.

## C    FAILURE CASES

Due to the reliance of our method on vanilla 3D Gaussian Splatting, which does not explicitly calculate lighting through normal vectors, combined with the random lighting conditions in different steps of the score distillation sampling, the lighting of objects in the scene is inconsistent. This inconsistency also contributes to the difficulty in generating semi-transparent materials. For instance, Figure 12 (a) illustrates the challenges with rendering glass and the highlights on its surface. Figure 12 (b) demonstrates another failure case involving the generation of text on the surface of 3D objects, which is attributed to the limited consistency of the diffusion models used for guidance.

## D    MORE ABLATIONS

We conduct comparative experiments on 100 scenes from the multiple objects subset of T3Bench He et al. (2023) and 30 scenes from our dataset, as detailed in Tables 2 and Table 3. Most scenes in our dataset include more than five objects, whereas the multiple objects subset of T3Bench typically includes only two objects. It can be observed that as the complexity of the scene increases, program-aided layout planning brings significant improvements.

Our task is to evaluate two complex 3D scenes that have been generated from the specific user prompt "{{user_prompt}}". I will provide you with images of these scenes, specifically image renderings, for each method used.

We want to assign a score from 1 to 100 (where 1 is the lowest and 100 is the highest) according to the provided four criteria:

**1. User Prompt & Scene Alignment:** Assess whether all objects mentioned in the user prompt "{{user_prompt}}" are present in the 3D scenes generated by both methods and whether the quantity of each type of object matches the numbers specified in the prompt. Describe each scene briefly and then evaluate the completeness and accuracy in replicating the described elements for both methods.

**2. Spatial Arrangement of Objects:** Look at the RGB images to assess the arrangement and positioning of objects within the scenes. Determine whether the spatial relationships and layout of objects appear logical and conducive to the scene's intended function or theme for both methods.

**3. Geometric Fidelity:** Examine each object within the scenes through the RGB images for both methods. Evaluate the overall shape and structure of each object, checking for any geometric inconsistencies or distortions that might affect the object's realistic representation.

**4. Overall Scene Quality:** Evaluate the overall coherence and technical quality of the scenes as a composite assessment, based on the integration of user prompt alignment, spatial arrangement, and geometric fidelity. Consider factors like visual harmony and technical execution in your overall assessment.

For each of the criteria, you will need to provide a score from 1 to 100 for each method. Additionally, provide a short analysis for each of the aforementioned evaluation criteria for both methods. The analysis should be very concise and accurate.

Let's step by step analyze the alignment of the scenes with the user prompt "{{user_prompt}}" and proceed to score and describe each method systematically.

# Example output:

**Analysis:**
**1. User Prompt & Scene Alignment:**
  - Method A: The scene includes objects such as trees, benches, and lamps; Score: 85
    All described objects are present, and the quantities are mostly accurate with minor deviations.
  - Method B: The scene includes the same objects but with slight variations in quantity; Score: 80
    Most objects are present, but there are notable discrepancies in object count.
**2. Spatial Arrangement of Objects:** ......
**3. Geometric Fidelity:** ......
**4. Overall Scene Quality:** ......
Final scores:
- Method A: 85, 78, 82, 90
- Method B: 80, 83, 88, 84

Figure 11: The prompt for guiding GPT-4 as a human-aligned evaluator

Table 4: Comparison of success rates with and without program-aided layout planning.

| Setup | Without program-aided | With program-aided |
|---|---|---|
| Success rate on our dataset | 30% | 77% |
| Success rate on the multiple objects subset of $T^3$Bench He et al. (2023) | 68% | 87% |

Ablation experiments are conducted on our dataset, as shown in Tables 2 and Table 3. Similar to the evaluation in Table 1, we calculate the CLIP score. Additionally, we provide GPT-4V with rendered images of the same 3D scene generated by different setups and require it to score each scene. The details of the evaluation are the same as those presented in Table 1 and are provided in Appendix A.3. Due to the randomness of GPT-4V and the different references used for scoring, the score of the "with both" setup is slightly different from that in Table 1.

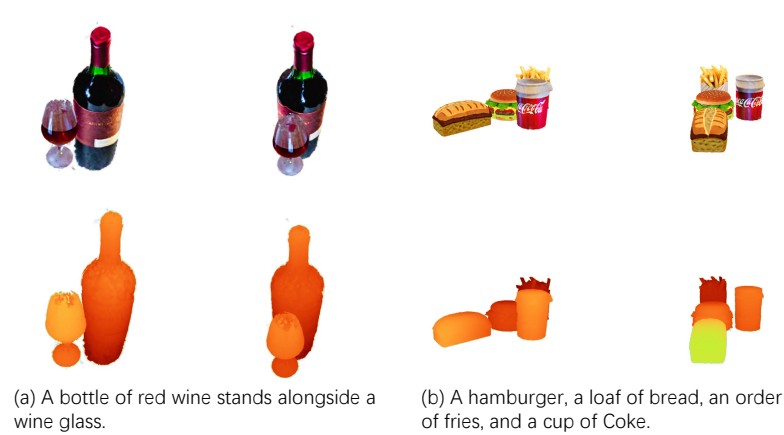

(a) A bottle of red wine stands alongside a wine glass.

(b) A hamburger, a loaf of bread, an order of fries, and a cup of Coke.

Figure 12: The rendered images and depth maps of failure cases of SEMANTICSDS

Without SemanticSDS, there is a significant decrease in CLIP score, prompt alignment, and scene quality. Without the object-specific view descriptor, there is a noticeable decrease in geometric fidelity and scene quality, with a slight decrease in prompt alignment.

Table 5: Quantitative comparison. Our SEMANTICSDS provides improved prompt alignment and the object-specific view descriptor helps achieve better geometric fidelity.

| Setup | Without SemanticSDS | Without object-specific view descriptor | With both |
|---|---|---|---|
| CLIP Score ↑ | 0.303 | 0.314 | 0.321 |
| Prompt Alignment ↑ | 82.4 | 86.6 | 90.2 |
| Spatial Arrangement ↑ | 80.9 | 79.6 | 83.5 |
| Geometric Fidelity ↑ | 80.1 | 77.4 | 84.7 |
| Scene Quality ↑ | 78.2 | 80.3 | 86.0 |

## E  EFFICIENCY ANALYSIS

Figure 13 illustrates the computational time breakdown for single and multiple objects generation scenarios using SEMANTICSDS. The visualization highlights three key components: rendering time, backward propagation time, and the time required to compute the SEMANTICSDS loss. It demonstrates that backward propagation time constitutes the majority of the total time. Even in scenarios involving multiple objects, the time spent on computing the SEMANTICSDS loss accounts for only about one-third of the total time. Furthermore, the application of SEMANTICSDS does not significantly increase the rendering time or the backpropagation time.

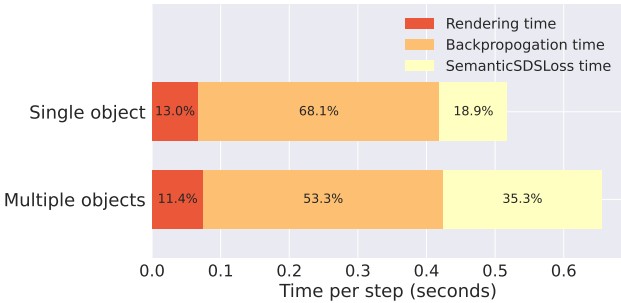

Figure 13: Efficiency analysis: Comparison of single vs. multiple objects generation using SEMANTICSDS. Percentages indicate the proportion of the total time spent on each component.

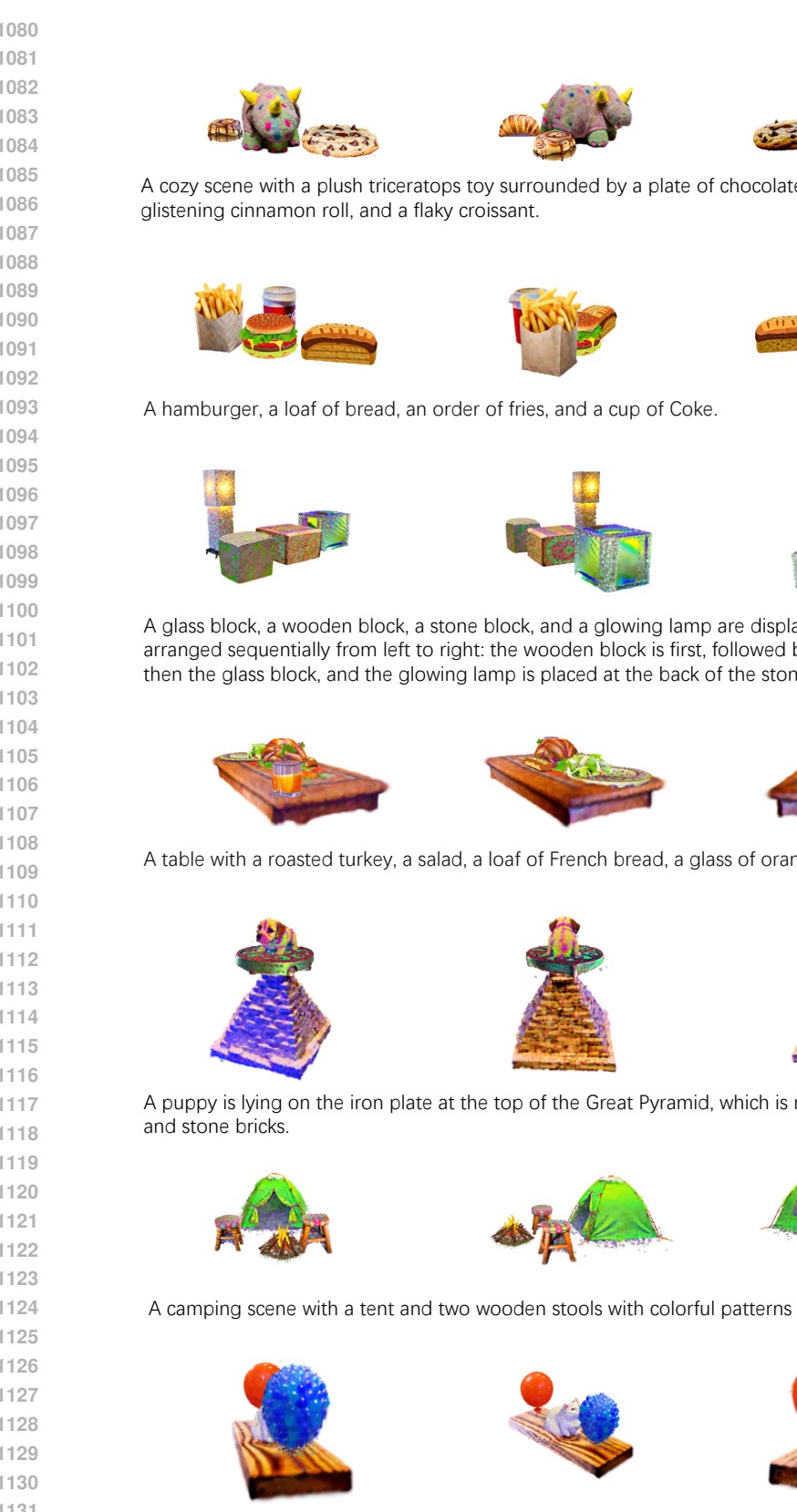

A cozy scene with a plush triceratops toy surrounded by a plate of chocolate chip cookies, a glistening cinnamon roll, and a flaky croissant.

A hamburger, a loaf of bread, an order of fries, and a cup of Coke.

A glass block, a wooden block, a stone block, and a glowing lamp are displayed. They are arranged sequentially from left to right: the wooden block is first, followed by the stone block, then the glass block, and the glowing lamp is placed at the back of the stone block.

A table with a roasted turkey, a salad, a loaf of French bread, a glass of orange juice and plate.

A puppy is lying on the iron plate at the top of the Great Pyramid, which is made of snow bricks and stone bricks.

A camping scene with a tent and two wooden stools with colorful patterns next to a campfire.

A white cat lies on a plank of wood, flanked by two sparkling balloons, one orange and one blue.

Figure 14: More synthesis results of multiple objects with our SEMANTICSDS.

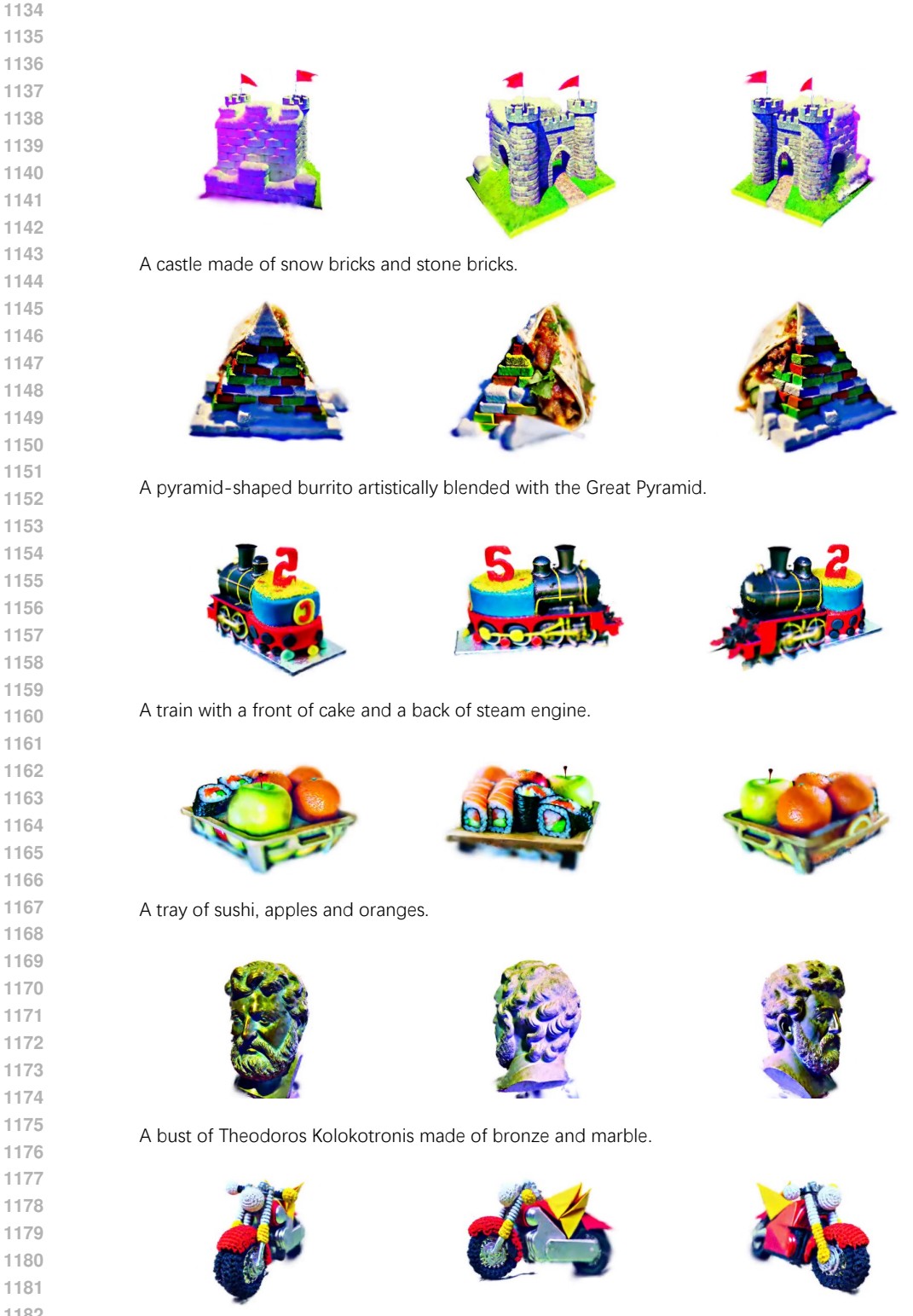

A castle made of snow bricks and stone bricks.

A pyramid-shaped burrito artistically blended with the Great Pyramid.

A train with a front of cake and a back of steam engine.

A tray of sushi, apples and oranges.

A bust of Theodoros Kolokotronis made of bronze and marble.

A motorcycle made of amigurumi and origami.

Figure 15: More synthesis results of single object with diverse attributes with our SEMANTICSDS.

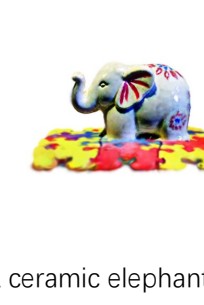 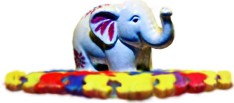 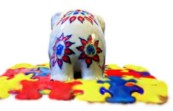

A ceramic elephant stands guard over an intricate puzzle yet to be completed.

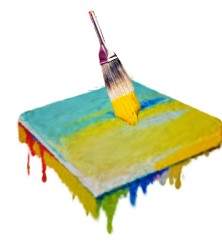 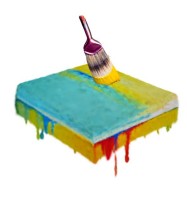 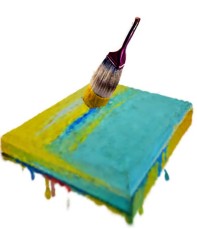

A dripping paintbrush stands poised above a half-finished canvas.

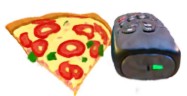 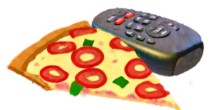 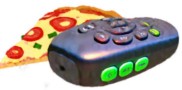

A half-eaten slice of pizza forgotten beside a remote control.

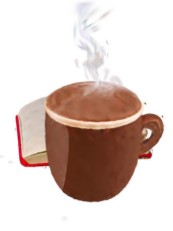 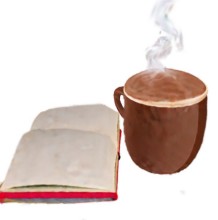 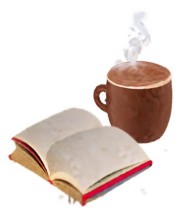

A thick novel is accompanied by a steaming cup of cocoa.

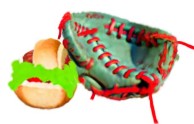 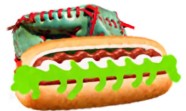 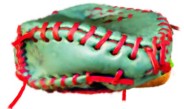

A baseball glove is forgotten next to a half-eaten hot dog.

Figure 16: Synthesis results on T³Bench (He et al., 2023) with our SEMANTICSDS.