# OpenReview forum: "Semantic Score Distillation Sampling for Compositional Text-to-3D Generation"
_ICLR.cc/2025/Conference — Submitted to ICLR 2025_

### Official Review · Reviewer_hCfH · 2024-10-27

**Soundness:** 3
**Presentation:** 3
**Contribution:** 2
**Rating:** 6
**Confidence:** 4

**Summary:**

This paper intruduce Semantic Score Distillation Sampling (SEMANTICSDS), a method that enhances the expressiveness and precision of compositional text-to-3D generation. It addresses the challenges of generating precise 3D layouts from vague language descriptions by leveraging program-aided layout planning, semantic embeddings, and explicit semantic guidance. SEMANTICSDS utilizes 3D Gaussian Splatting (3DGS) as the 3D representation and consists of three key steps: program-aided layout planning, semantic embeddings, and rendering semantic maps for fine-grained optimization and compositional generation. The paper demonstrates that SEMANTICSDS achieves state-of-the-art results in generating complex 3D content.

**Strengths:**

1. SEMANTICSDS demonstrates the ability to generate and manage complex inter-object relationships effectively, such as arranging objects with multiple, overlapping attributes (e.g., a “car made of sushi and LEGO”). This makes it a strong candidate for detailed scene design tasks.
2.

**Weaknesses:**

1. The Semantic 3D Gaussian Representation proposed in this paper is similar to the approach used in LangSplat. However, the paper does not adequately discuss or clarify the differences between the two approaches.
2. The experimental section is insufficient; the authors are encouraged to incorporate results using LucidDreamer’s ISM optimization to ensure a fair comparison and provide a more comprehensive evaluation of the method’s performance.

**Questions:**

1. Could the authors provide more details about the computational resources required for training, such as GPU specifications, memory usage, and total computational cost? This would help clarify the practical feasibility of the proposed method.
2. As the number of objects in a scene increases, does the training time scale linearly? If not, could the authors elaborate on how the system handles the increased complexity and whether there are any bottlenecks or optimizations to manage training efficiency?

---

> ### Author Response · Authors · 2024-11-22
> **Response to Reviewer hCfH - Part 1**
>
> *We sincerely thank you for your time and efforts in reviewing our paper and for your valuable feedback. We are glad to see that our framework is effective and strong. Please see below for our detailed responses to your comments. The revised texts in the updated paper are denoted in blue.*
>
> **Q1: The Semantic 3D Gaussian Representation proposed in this paper is similar to the approach used in LangSplat. However, the paper does not adequately discuss or clarify the differences between the two approaches.**
>
> A1: Thank you for your suggestions. We would like to highlight four main differences between our framework and LangSplat:
>
> * **Guidance for Semantic 3D Gaussian Representation**: In LangSplat, the semantic 3D Gaussian representation is optimized using images processed through Segment Anything (SAM) and CLIP. In contrast, our SemanticSDS employs a program-aided layout planning method to associate semantic 3D Gaussian representations with precise 3D positions and relationships.
> * **Loss Functions for Training Autoencoders**: LangSplat uses L1 and cosine distance loss. In contrast, we use L1 loss and the symmetric cross-entropy loss from CLIP. This is because we need to convert semantic maps into 0-1 semantic masks, and the symmetric cross-entropy loss is more suitable for classification.
> * **Utilization of Semantic 3D Gaussian Representation**: In LangSplat, semantic 3D Gaussian representations are only used for semantic alignment. Conversely, we utilize semantic 3D Gaussian representations to render a semantic map and propose a novel region-specific SDS to enable fine-grained 3D manipulation and optimization.
> * **Applied Scenario**: LangSplat is applied in 3D object localization and semantic segmentation, which are understanding tasks. Our SemanticSDS is applied in compositional text-to-3D generation, which involves creative and generative tasks.
>
> **Q2: The experimental section is insufficient; the authors are encouraged to incorporate results using LucidDreamer’s ISM optimization to ensure a fair comparison and provide a more comprehensive evaluation of the method’s performance.**
>
> A2: Thank you for your constructive suggestion. LucidDreamer demonstrates that ISM performs better than vanilla SDS. However, the main baselines, GALA3D and GSGEN, do not use LucidDreamer’s ISM, so not using ISM ensures a fairer comparison. Nonetheless, your suggestion is valuable. Our framework focuses on improving attribute and spatial alignment and can be combined with methods like LucidDreamer for higher fidelity. We plan to integrate ISM into our framework in the future.
>
> We conducted comparative experiments on 100 scenes from the multiple objects subset of T3Bench \[1\] and 30 scenes from our dataset, detailed in Table 2 and Table 3\. Most scenes in our dataset include more than 5 objects, while the multiple objects subset of T3Bench often includes only 2 objects. It can be observed that as the complexity of the scene increases, program-aided layout planning brings significant improvement.
>
> | Setup | Without program-aided | With program-aided |
> | ----- | ----- | ----- |
> | Success rate on our dataset detailed in Table 2 and Table 3 | 30% | 77% |
> | Success rate on T3Bench \[1\] | 68% | 87% |
>
> We conducted ablation experiments on our dataset shown in Table 2 and Table 3\. Same as evaluation in Table 1, we calculated the CLIP score. We also provided GPT-4V with rendered images of the same 3D scene generated by different setups and required it to score each scene. The details of evaluation are the same as in Table 1, presented in Appendix A.3. Due to the randomness of GPT-4V and different references used during scoring, the score of the "with both" setup is slightly different from the one in Table 1\. Without SemanticSDS, there is a significant drop in CLIP score, prompt alignment, and scene quality. Without object-specific view descriptor, there is a notable decrease in geometric fidelity and scene quality, and a slight decrease in prompt alignment.
>
> | Setup | Without SemanticSDS | Without object-specific view descriptor | With both |
> | ----- | ----- | ----- | ----- |
> | CLIP Score ↑ | 0.303 | 0.314 | 0.321 |
> | Prompt Alignment ↑ | 82.4 | 86.6 | 90.2 |
> | Spatial Arrangement ↑ | 80.9 | 79.6 | 83.5 |
> | Geometric Fidelity ↑ | 80.1 | 77.4 | 84.7 |
> | Scene Quality ↑ | 78.2 | 80.3 | 86.0 |
>
> The above quantitative ablation experiments validate the effectiveness of our designs. Thank you for your valuable suggestion again. **We have added these experimental results to Appendix D, highlighted in blue.**
>
> \[1\] He, Yuze, et al. "T ^ 3 Bench: Benchmarking Current Progress in Text-to-3D Generation." *arXiv preprint arXiv:2310.02977* (2023).

---

> ### Author Response · Authors · 2024-11-22
> **Response to Reviewer hCfH - Part 2**
>
> **Q3: Could the authors provide more details about the computational resources required for training, such as GPU specifications, memory usage, and total computational cost? This would help clarify the practical feasibility of the proposed method.**
>
> A3: Thank you for your question. We conducted our experiments on a single A100-PCIE-40GB GPU. Compared to GSGEN, which serves as the codebase for our method, we used the same hyperparameters, such as 4096 initial Gaussians for each object. Our method showed only a 21% increase in memory usage primarily due to the higher number of initial Gaussians required for generating complex scenes involving multiple objects. The training time for our method was only 27% longer than that of GSGEN.
>
> | Method | GSGEN | Ours |
> | ----- | ----- | ----- |
> | Memory usage | 10.6GB | 12.9GB |
> | Training time | 2.2 hours | 2.85 hours   |
>
>
> **Q4: As the number of objects in a scene increases, does the training time scale linearly? If not, could the authors elaborate on how the system handles the increased complexity and whether there are any bottlenecks or optimizations to manage training efficiency?**
>
> A4: As the number of objects in a scene increases, the training time does not scale linearly. Thank you for your constructive suggestion. **We have added Section E and Figure 13 in the revised manuscript to address efficiency analysis.** As illustrated in Figure 13, in scenarios involving multiple objects, the time spent on computing the semanticSDS loss accounts for only about one-third of the total time per training step. And it indicates that the backpropagation time constitutes the majority of the total time, which is not significantly affected by our method.
>
> Our training alternates between local and global optimization. During local optimization, we render only a single object, making other objects invisible. Consequently, most subprompts correspond to semantic masks that are entirely zero, and thus, we don't predict the denoising scores for them to handle the increased complexity.
>
> Additionally, in the differentiable rasterizer for Gaussians \[1\], sorting Gaussians for an entire image for alpha-blending is the primary time-consuming process. In our implementation, rendering both RGB images and semantic maps requires sorting Gaussians only once, the same as rendering just the RGB image. Therefore, the rendering time does not increase significantly, as shown in the added Figure 13\.
>
> \[1\] Kerbl, Bernhard, et al. "3D Gaussian Splatting for Real-Time Radiance Field Rendering." *ACM Trans. Graph.* 42.4 (2023): 139-1.

---

### Official Review · Reviewer_3TfJ · 2024-11-03

**Soundness:** 2
**Presentation:** 2
**Contribution:** 3
**Rating:** 6
**Confidence:** 4

**Summary:**

This paper presents a semantic score distillation sampling method for compositional text-to-3D generation by exploiting semantic embeddings and region-specific SDS. Experiments demonstrate promising results of the proposed model for generating complex 3D content.

**Strengths:**

++ The main idea is novel and interesting for the emerging topic of compositional text-to-3D generation.

++ Clear performance boosts are attained in experimental sections.

**Weaknesses:**

-- No information of testing data for evaluation is provided. Moreover, it is necessary to introduce the selected 30 scenes for user study. I am happy to see more convincing results or user study over larger-scale test samples (more than 100 scenes at least).

-- In section 5.2, only qualitative comparisons are shown to validate the effectiveness of each design. It is better to perform quantitative comparison on testing data for evaluation.

-- T3Bench [A] contains the subset of multiple objects, and I am curious to see more evaluation over this widely used benchmark.

[A] He Y, Bai Y, Lin M, et al. T $^ 3$ Bench: Benchmarking Current Progress in Text-to-3D Generation[J]. arXiv preprint arXiv:2310.02977, 2023.

**Questions:**

I will increase my rating if more convincing experimental results are provided.

---

> ### Author Response · Authors · 2024-11-22
> **Response to Reviewer 3TfJ**
>
> *We sincerely thank you for your time and efforts in reviewing our paper and for your valuable feedback. We are glad to see that our idea is novel and interesting, and the performance is promising. Please see below for our detailed responses to your comments. The revised texts in the updated paper are denoted in blue.*
>
> **Q1: No information on testing data for evaluation is provided. Moreover, it is necessary to introduce the selected 30 scenes for user study.**
>
> A1: Thank you for pointing out this. We utilized the same dataset for quantitative comparisons in Table 1 as we did in the user study. This dataset, which includes 30 scenes involving about 160 objects, is detailed in **the newly added Table 2 and Table 3 in Appendix A.3**.  The objects exhibit varied attributes and possess complex relative positions, making them suitable for evaluating intricate 3D scenes.
>
> **Q2: In section 5.2, only qualitative comparisons are shown to validate the effectiveness of each design. It is better to perform the quantitative comparison on testing data for evaluation.**
>
> A2: Thank you for your constructive suggestion. We conducted comparative experiments on 100 scenes from the multiple objects subset of T3Bench \[1\] and 30 scenes from our dataset, detailed in Table 2 and Table 3\. Most scenes in our dataset include more than 5 objects, while the multiple objects subset of T3Bench often includes only 2 objects. It can be observed that as the complexity of the scene increases, program-aided layout planning brings significant improvement.
>
> | Setup | Without program-aided | With program-aided |
> | ----- | ----- | ----- |
> | Success rate on our dataset is detailed in Table 2 and Table 3 | 30% | 77% |
> | Success rate on T3Bench \[1\] | 68% | 87% |
>
> We conducted ablation experiments on our dataset shown in Table 2 and Table 3\. Same as the evaluation in Table 1, we calculated the CLIP score. We also provided GPT-4V with rendered images of the same 3D scene generated by different setups and required it to score each scene. The details of the evaluation are the same as in Table 1, presented in Appendix A.3. Due to the randomness of GPT-4V and different references used during scoring, the score of the "with both" setup is slightly different from the one in Table 1\. Without SemanticSDS, there is a significant drop in CLIP score, prompt alignment, and scene quality. Without an object-specific view descriptor, there is a notable decrease in geometric fidelity and scene quality and a slight decrease in prompt alignment.
>
> | Setup | Without SemanticSDS | Without object-specific view descriptor | With both |
> | ----- | ----- | ----- | ----- |
> | CLIP Score ↑ | 0.303 | 0.314 | 0.321 |
> | Prompt Alignment ↑ | 82.4 | 86.6 | 90.2 |
> | Spatial Arrangement ↑ | 80.9 | 79.6 | 83.5 |
> | Geometric Fidelity ↑ | 80.1 | 77.4 | 84.7 |
> | Scene Quality ↑ | 78.2 | 80.3 | 86.0 |
>
> The above quantitative ablation experiments validate the effectiveness of our designs. Thank you for your valuable suggestion again. **We have added these experimental results to Appendix D, highlighted in blue.**
>
> \[1\] He, Yuze, et al. "T $^ 3$ Bench: Benchmarking Current Progress in Text-to-3D Generation." *arXiv preprint arXiv:2310.02977* (2023).
>
> **Q3: I am happy to see more convincing results or user study over larger-scale test samples (more than 100 scenes at least). T3Bench contains a subset of multiple objects, and I am curious to see more evaluation of this widely used benchmark.**
>
> A3: It is important to emphasize that most scenes in our dataset include more than 5 objects, while the multiple objects subset of T3Bench often includes only 2 objects. In our dataset, the objects exhibit varied attributes and possess complex relative positions, making them more suitable for evaluating the generation of intricate 3D scenes involving multiple objects with varied attributes. The table "Comparison of success rates with and without program-aided layout planning" in Question 2 can substantiate this point. On the multiple objects subset of T3Bench, without program-aided layout planning, a decent success rate can still be achieved. Besides, we also provide visual synthesis results on T3Bench in the **newly added Figure 16\.**

---

> ### Author Response · Authors · 2024-11-25
> **Gentle Reminder**
>
> Dear reviewer 3TfJ:
>
> We sincerely appreciate the time and effort you dedicated to reviewing our paper. In response to your concerns, we have conducted additional experiments and provided an in-depth analysis to demonstrate the superiority of our proposed SemanticSDS during the discussion period.
>
> As the discussion period concludes in two days, we kindly request, if possible, that you review our rebuttal at your convenience. Should there be any further points requiring clarification or improvement, please know that we are fully committed to addressing them promptly. Thank you once again for your invaluable contribution to our research.
>
> Warm regards,
>
> The Authors

---

> > ### Comment · Reviewer_3TfJ · 2024-11-26
> > **Thanks for the response**
> >
> > Most of my concerns have been addressed in the author responses. Thus I have increased my rating.

---

> > > ### Author Response · Authors · 2024-11-26
> > > **Thank you for your support**
> > >
> > > Dear Reviewer 3TfJ，
> > >
> > > Thank you for raising score! We sincerely appreciate your valuable comments and your precious time in reviewing our paper!
> > >
> > > Warm Regards,
> > >
> > > The Authors

---

### Official Review · Reviewer_rjrF · 2024-11-06

**Soundness:** 3
**Presentation:** 3
**Contribution:** 3
**Rating:** 6
**Confidence:** 4

**Summary:**

This paper presents a novel approach, Semantic Score Distillation Sampling (SEMANTICSDS), aimed at enhancing compositional text-to-3D generation by integrating explicit semantic guidance. SEMANTICSDS focuses on fine-grained control over complex scenes by introducing program-aided layout planning, semantic maps, and object-specific view descriptors, enhancing the expressiveness and accuracy of 3D generation for complex objects and scenes.

**Strengths:**

+ The SEMANTICSDS framework is highly effective in generating complex 3D content.
+ Program-aided layout planning is a notable addition, offering structured scene arrangements.
+ The introduction of semantic maps and region-specific SDS improves control and precision in generating complex scenes with multiple attribute

**Weaknesses:**

- The generated scene textures lack realism, which may limit visual appeal and applicability.
- The paper does not showcase any failure cases, which could help illustrate method limitations.
- Quantitative comparisons in the experiments section lack details on specific datasets used for evaluation.

**Questions:**

1. How does SEMANTICSDS perform with prompts involving highly abstract concepts or non-physical attributes?
2. How does the framework handle scenes with overlapping objects or closely packed spatial arrangements?
3. Program-aided layout planning is interesting; how is the reliability and accuracy of the program ensured?
4. What is the success rate of the LLM?
5. How long does it take to generate a scene, and what is the estimated time per step?
6. In Table 1, how large is the dataset used for qualitative comparison, and how was it constructed?

---

> ### Author Response · Authors · 2024-11-22
> **Response to Reviewer rjrF - Part 1**
>
> *We sincerely thank you for your time and efforts in reviewing our paper and for your valuable feedback. We are glad to see that our framework is highly effective and produces notable additions to complex scene generation. Please see below for our detailed responses to your comments. The revised texts in the updated paper are denoted in blue.*
>
> **Q1: The generated scene textures lack realism, which may limit visual appeal and applicability.**
>
> A1: The key contributions of our method, including program-aided layout planning, semanticSDS, and the object-specific view descriptor, can be applied to other text-to-single object methods to generate complex 3D scenes with multiple objects and various attributes. Our framework focuses on improving attribute and spatial alignment of compositional generation, which can be used alongside methods focused on texture realism such as Interval Score Matching \[1\] and Bootstrapped Score Distillation \[2\].
>
> \[1\]  Liang, Yixun, et al. "Luciddreamer: Towards high-fidelity text-to-3d generation via interval score matching." *Proceedings of the IEEE/CVF Conference on Computer Vision and Pattern Recognition*. 2024\.
>
> \[2\] Sun, Jingxiang, et al. "Dreamcraft3d: Hierarchical 3d generation with bootstrapped diffusion prior." *arXiv preprint arXiv:2310.16818* (2023).
>
> **Q2: The paper does not showcase any failure cases, which could help illustrate method limitations.**
>
> A2: Thank you for your constructive suggestion. We have **added section C and Figure 12** in the Appendix, with revised sections highlighted in blue. Figure 12 shows the failure cases of our method on light consistency and generating text on the surface of objects.
>
> **Q3: Quantitative comparisons in the experiments section lack details on specific datasets used for evaluation. In Table 1, how large is the dataset used for qualitative comparison, and how was it constructed?**
>
> A3: Thank you for pointing this out. We utilized the same dataset for quantitative comparisons in Table 1 as we did in the user study. This dataset, which includes about 160 objects, is detailed in **the newly added Table 2 and Table 3 in Appendix A.3**.  The objects exhibit varied attributes and possess complex relative positions, making it suitable for evaluating intricate 3D scenes.
>
> The dataset is constructed by asking LLMs to act as a professional set designer to design objects that logically form an interesting and reasonable scene. The scene description includes the relative positioning of the objects to each other.
>
> **Q4: How does SEMANTICSDS perform with prompts involving highly abstract concepts or non-physical attributes?**
>
> A4: Generating highly abstract concepts or non-physical attributes in text-to-3D is an ongoing exploration. SemanticSDS unlocks the compositional capabilities of existing pre-trained diffusion models, potentially addressing this issue. An example in the newly added Figure 16 presents the “steaming” attribute for the scene “A thick novel is accompanied by a steaming cup of cocoa.”
>
> **Q5: How does the framework handle scenes with overlapping objects or closely packed spatial arrangements?**
>
> A5: Thank you for your insightful comment. When objects overlap, the semantic embeddings of the Gaussians in the overlapped region will be assigned the weighted average of the semantic embeddings of the relevant objects. The weights depend on the distance from these Gaussians to the center of each relevant object. This approach ensures that the semantic maps smoothly transition from one object to the other in the overlapped region.
>
> For closely packed spatial arrangements, it is worth exploring more fine-grained semantic embeddings optimized based on attention maps from 2D diffusion models pre-trained on large-scale 2D image datasets. Regions in the attention maps with high similarity to the subprompts can serve as estimates for the ground truth semantic maps. Thank you again for your insightful comment. We will consider this promising direction in our future work.

---

> ### Author Response · Authors · 2024-11-22
> **Response to Reviewer rjrF - Part 2**
>
> **Q6: Program-aided layout planning is interesting; how is the reliability and accuracy of the program ensured?**
>
> A6: As shown in Figures 8 and 9, we employ chain of thoughts\[1\] and in-context learning\[2\] to drive the LLM to generate the program, improving the reliability and accuracy of the program. We conduct experiments on 100 scenes from the multiple objects subset of T3Bench [3], and successfully run programs for 94 scenes.
>
> \[1\] Wei, Jason, et al. "Chain-of-thought prompting elicits reasoning in large language models." *Advances in neural information processing systems* 35 (2022): 24824-24837.
>
> \[2\] Dong, Qingxiu, et al. "A survey on in-context learning." *arXiv preprint arXiv:2301.00234* (2022).
>
> \[3\] He, Yuze, et al. "T $^ 3$ Bench: Benchmarking Current Progress in Text-to-3D Generation." *arXiv preprint arXiv:2310.02977* (2023).
>
> **Q7: What is the success rate of the LLM?**
>
> A7: Thank you for your valuable inquiry. Using the code provided in the supplementary material, layouts can be generated in batches and visualized. The visualization for each scene is a video with the viewpoint rotating around the scene center. The frames of the videos are similar to Figure 6 in our paper. We conduct comparative experiments on 100 scenes from the multiple objects subset of T3Bench \[3\] and 30 scenes from our dataset detailed in Table 2 and Table 3\. Most scenes in our dataset include more than 5 objects, while the multiple objects subset of T3Bench often includes only 2 objects. It can be observed that as the complexity of the scene increases, program-aided layout planning brings significant improvement.
>
> | Setup | Without program-aided | With program-aided |
> | ----- | ----- | ----- |
> | Success rate on our dataset detailed in Table 2 and Table 3 | 30% | 77% |
> | Success rate on T3Bench \[3\] | 68% | 87% |
>
> \[3\] He, Yuze, et al. "T $^ 3$ Bench: Benchmarking Current Progress in Text-to-3D Generation." *arXiv preprint arXiv:2310.02977* (2023).
>
> **Q8: How long does it take to generate a scene, and what is the estimated time per step?**
>
> A8: Thank you for your question. We conducted our experiments on a single A100-PCIE-40GB GPU. Compared to GSGEN, which serves as the codebase for our method, we used the same hyperparameters, such as 4096 initial Gaussians for each object. The training time for our method was only 27% longer than that of GSGEN.
>
> | Method | GSGEN | Ours |
> | ----- | ----- | ----- |
> | Training time | 2.2 hours | 2.85 hours |
>
> Additionally, We have added Section E and Figure 13 in the revised manuscript to address efficiency analysis. The estimated time per step is about 0.656 seconds. As illustrated in Figure 13, in scenarios involving multiple objects, the time spent on computing the semanticSDS loss accounts for only about one-third of the total time per training step. And it indicates that the back-propagation time constitutes the majority of the total time, which is not significantly affected by our method.

---

> > ### Comment · Reviewer_rjrF · 2024-11-27
> > **Thanks for your response**
> >
> > The authors have addressed my concerns.

---

### Official Review · Reviewer_Pn2r · 2024-11-06

**Soundness:** 3
**Presentation:** 2
**Contribution:** 3
**Rating:** 5
**Confidence:** 4

**Summary:**

This paper introduces a score distillation-based learning framework for compositional text-to-3D generation. This framework consists of two components: (1) Program-aided layout planning; (2) Semantic SDS. The layout planning module utilizes LLMs to produce layout-related python codes, then the layouts of different objects can be computed by executing python codes. In this way the hallucinations of LLMs in numerical computation can be alleviated. Semantic SDS uses semantic maps rendered from 3DGS as guidance to merge multiple denoising scores. For experiments, qualitative and quantitative evaluations are performed to demonstrate the superiority of the proposed method over previous methods: GraphDreamer, LucidDreamer, GALA3D, GSGEN, while ablation analysis demonstrates the effectiveness of each component.

**Strengths:**

1. Although semantic 3DGS is not rare, this work combines semantic maps rendering and score distillation to improve compositional text-to-3D generation, which is worthy of recognition in terms of reliability and soundness.
2. The proposed method shows satisfactory performance compared with previous methods, especially in attribute and spatial alignment. The evaluation is thorough, including qualitative and quantitative comparisons and user studies.
3. The design of Object-Specific View Descriptor is well-motivated, and I particularly appreciate the presentation in Figure 3.

**Weaknesses:**

1. Overall, this work proposes some incremental improvements, for example, semantic score distillation is a very direct combination of semantic 3DGS and score distillation (i.e., a SDS loss weighted by semantic probability maps). Experimental evaluation shows that the proposed method outperforms previous evaluation methods, but not significantly. For instance, in Figure 4, the compared methods GALA3D and GSGEN have higher fidelity and less noise and artifacts.
2. Compared with the baseline GSGEN, SemanticSDS causes quite severe blurring, artifacts, and noise, as shown in Figure 4. Although SemanticSDS effectively improves the spatial layout and attribute binding, its degradation of visual quality is not negligible.
3. Some details of SemanticSDS seem to be missing. For example, I know that 3D Gaussians are initialized from Shap-E, but how is the correspondence between these 3D Gaussians and subprompts obtained? Also, are the semantic embeddings of 3D Gaussians optimized during training?
4. Based on Eq.(9), it seems that k * l denoising scores need to be predicted, which brings great computational and time costs. Efficiency analysis and comparison should be provided in this case.

**Questions:**

1. How is the correspondence between initial 3D Gaussians and subprompts obtained? Also, are the semantic embeddings of 3D Gaussians optimized during training?
2. In Figure 2, why is there noise at the edges of objects in semantic maps?

---

> ### Author Response · Authors · 2024-11-22
> **Response to Reviewer Pn2r - Part 1**
>
> *We sincerely thank you for your time and efforts in reviewing our paper and for your valuable feedback. We are glad to see that our method has been recognized for its improvements in attribute and spatial alignment. Please see below for our detailed responses to your comments. The revised texts in the updated paper are denoted in blue.*
>
> **Q1: Compared with the baseline GSGEN, SemanticSDS causes quite severe blurring, artifacts, and noise, as shown in Figure 4\. Although SemanticSDS effectively improves the spatial layout and attribute binding, its degradation of visual quality is not negligible.**
>
> A1: Thank you for your observation. In all experiments, both SemanticSDS and GSGEN were optimized for 15000 steps. However, SemanticSDS optimizes multiple objects simultaneously, while GSGEN typically optimizes only one object. This difference leads to a decrease in optimization steps per object, resulting in artifacts. In scenarios with fewer objects, such as those in Figure 15 and the newly added Figure 16, the blurring and artifacts are significantly reduced. Finally, we want to emphasize that the main focus of our framework is to improve the attribute and spatial alignment of compositional generation. Increasing the number of optimization steps could potentially mitigate the blurring and artifacts; however, this would come at the cost of increased optimization time. Therefore, we did not specifically adjust the parameters for the generation of each object, nor did we cherry-pick the results displayed.
>
> **Q2: How is the correspondence between initial 3D Gaussians and subprompts obtained?**
>
> A2: During the program-aided layout planning, the layout space for object $O\_k$ is decomposed into $n\_k$ complementary 3D bounding boxes, each associated with different subprompts. These 3D bounding boxes are under a normalized coordinate system, where the coordinates range between 0 and 1\.
>
> For each object $O\_k$, we utilize Shap-E to generate the positions of Gaussians based on the corresponding prompt $y\_k$. After transforming the Gaussians into the normalized coordinate system, we enhance the Gaussians within the 3D bounding boxes corresponding to subprompt $y\_{k,l}$ with semantic embeddings $E(\\Phi(y\_{k,l}))$. Here, $\\Phi(\\cdot)$ is the CLIP text encoder and $E(\\cdot)$ is the encoder of our trained autoencoder. Subsequently, we transform the Gaussians to global coordinates using the scale factors, Euler angles, and translation vectors specified in the layout for object $O\_k$.
>
> Thank you for pointing out the unclear parts of our paper. We have updated the manuscript to provide a clearer explanation. The revised sections are highlighted in blue.
>
> **Q3: Are the semantic embeddings of 3D Gaussians optimized during training?**
>
> A3: The semantic embeddings of 3D Gaussians are not optimized during training. Obtaining the ground truth for optimizing semantic embeddings is a challenging task. In the early stages of training, the rendered RGB images are highly noisy, making it difficult to obtain accurate ground truth semantic maps. In the later stages of training, the primary geometry and appearance of objects are already determined, and optimizing the semantic embeddings at this point yields limited improvements. Additionally, optimizing the 3D objects themselves based on the rendered RGB images from potentially erroneous 3D objects might not be effective.
>
> Since our goal is to ensure that the generated scene aligns well with the input prompt, we use layouts that are semantically decomposed from the input prompt to assign values to the semantic embeddings of 3D Gaussians.
>
> However, optimizing the semantic embeddings during training is indeed a promising direction. For example, it is worth exploring the optimization of semantic embeddings based on attention maps from 2D diffusion models pre-trained on large-scale 2D image datasets. Regions in the attention maps with high similarity to the subprompts can serve as estimates for the ground truth semantic maps.
>
> Thank you again for your insightful comment. We will consider this direction in our future work.

---

> ### Author Response · Authors · 2024-11-22
> **Response to Reviewer Pn2r - Part 2**
>
> **Q4: Overall, this work proposes some incremental improvements, for example, semantic score distillation is a very direct combination of semantic 3DGS and score distillation (i.e., a SDS loss weighted by semantic probability maps).**
>
> A4: The primary motivation behind SemanticSDS is to generate intricate and complex 3D scenes with multiple objects, each possessing distinct attributes. Achieving precise control over optimizing different parts of objects with distinct attributes is a significant challenge that cannot be addressed by merely combining existing techniques. A naive combination often leads to severe attribute mixing or pattern leaking when conditioning the diffusion models on a single text prompt for fine-grained object generation.
>
> After achieving fine control over the generation of single objects, we introduce program-aided layout planning to accurately determine the relative positions of multiple objects and an object-specific view descriptor for stable generation. These are unique contributions of our work and systematically address the challenges faced when generating intricate and complex 3D scenes.
>
> **Q5: Based on Eq.(9), it seems that k \* l denoising scores need to be predicted, which brings great computational and time costs. Efficiency analysis and comparison should be provided in this case.**
>
> A5: Thank you for your constructive suggestion. **We have added Section E and Figure 13 in the revised manuscript to address efficiency analysis.** As illustrated in **Figure 13**, in scenarios involving multiple objects, the time spent on computing the semanticSDS loss accounts for only about one-third of the total time per training step. And it indicates that the backpropagation time constitutes the majority of the total time, which is not significantly affected by our method.
>
> It is important to note that our training alternates between local and global optimization. During local optimization, we render only a single object, making other objects invisible. Consequently, most subprompts correspond to semantic masks that are entirely zero, and thus, predicting the denoising scores for them is unnecessary. Therefore, it is only during global optimization, and specifically when all subprompts correspond to unoccluded regions, that predicting all denoising scores is required.
> Additionally, we conducted our experiments on a single A100-PCIE-40GB GPU. Compared to GSGEN, which serves as the codebase for our method, we used the same hyperparameters, such as 4096 initial Gaussians for each object. Our method showed only a 21% increase in memory usage primarily due to the higher number of initial Gaussians required for generating complex scenes involving multiple objects. The training time for our method was only 27% longer than that of GSGEN.
>
> | Method | GSGEN | Ours |
> | ----- | ----- | ----- |
> | Memory usage | 10.6GB | 12.9GB |
> | Training time | 2.2 hours | 2.85 hours   |
>
> **Q6: In Figure 2, why is there noise at the edges of objects in semantic maps?**
>
> A6: Thank you for pointing out our oversight. We examine the visualized semantic maps saved in two different ways during training: one saved locally as mp4 files capturing the semantic maps from surrounding scenes, and the other uploaded to wandb (Weights & Biases) as gif files. We find noise at the edges of objects only in the mp4 files. After modifying the visualization method, the noise disappears. To further verify, we visualize the semantic masks derived from the semantic maps, and the semantic masks cover the edges of objects. The visualization of semantic masks has been added in the updated Appendix A.2 as Figure 10\. We have also replaced the original semantic maps in Figure 2 with the correctly visualized ones. Thank you again for your meticulous review.

---

> ### Author Response · Authors · 2024-11-25
> **Gentle Reminder**
>
> Dear reviewer Pn2r:
>
> We sincerely appreciate the time and effort you dedicated to reviewing our paper. In response to your concerns, we have conducted additional experiments and provided an in-depth analysis to demonstrate both effectiveness and efficiency of our proposed SemanticSDS during the discussion period.
>
> As the discussion period concludes in two days, we kindly request, if possible, that you review our rebuttal at your convenience. Should there be any further points requiring clarification or improvement, please know that we are fully committed to addressing them promptly. Thank you once again for your invaluable contribution to our research.
>
> Warm regards,
>
> The Authors

---

> > ### Comment · Reviewer_Pn2r · 2024-11-26
> >
> > Thanks the authors for the detailed response. Some of my questions as in Q2, Q3, and Q6 have been well answered. However, there are still some concerns:
> >
> > Q1. The authors argue that artifacts and blurring in the results compared to GSGEN are mainly caused by the limited number of iterations for multiple objects. I think a strong evidence for this would be to further increase the number of iterations for the examples in Figure 4, achieving better visual quality comparable to GSGEN, rather than providing extra examples as in Figure 15 and Figure 16 without comparison with GSGEN. Even for the results in Figures 15 and 16, the artifacts and blurring are still obvious.
> >
> > Q4. I agree with the authors that complex 3D scene generation is significantly challenging. However, the technical contributions of this work are fairly straightforward and not very inspiring. First, the accurate layout of 3D scene is largely due to the planning ability of LLM. The proposed program-aided planning does reduce the hallucination but is somewhat engineering. Second, semantic SDS directly utilizes semantic masks for SDS, which may be able to maintain the semantics from the initial layout, but the visual quality is not satisfactory. As shown in Figure 7, for "a house that is half made of LEGO and half of snow", the area where the attributes change is noisy.
> >
> > Q5. Alternating training involves a tradeoff between local and global optimization. An analysis of this is important but not provided. For example, what are the results of "local optimization only" and "global optimization only"? And what is the influence of local-global ratio during alternating training?
> >
> > Thanks again to the authors for their time and effort for the detailed response. Unfortunately, given that the above concerns remain and that are unlikely to be addressed in this rebuttal, I will keep my original rating.

---

> ### Author Response · Authors · 2024-11-26
> **Response to reviewer Pn2r**
>
> We sincerely thank you for your feedback and discussion. In response to your remaining concerns, we further make clarification of these questions:
>
> **Q1. The authors argue that artifacts and blurring in the results compared to GSGEN are mainly caused by the limited number of iterations for multiple objects. I think a strong evidence for this would be to further increase the number of iterations for the examples in Figure 4, achieving better visual quality comparable to GSGEN, rather than providing extra examples as in Figure 15 and Figure 16 without comparison with GSGEN. Even for the results in Figures 15 and 16, the artifacts and blurring are still obvious.**
>
> A1: We want to emphasize that in terms of optimizing visual quality, the optimal number of iterations varies significantly across different objects, and so do the optimization hyperparameters for each object. Unlike GSGEN, which typically focuses on optimizing a single object, our SemanticSDS is specifically designed for the compositional generation of multiple objects or attributes. We have showcased numerous visual results to demonstrate our system's superiority in compositional generation while maintaining satisfactory visual quality. Theoretically, it is possible to optimize each object separately with different iterations and hyperparameters to achieve the best generation results. However, this falls outside our primary focus. We appreciate your detailed feedback and kindly ask for your understanding of our main focus and novel contributions to compositional generation.
>
> **Q4. I agree with the authors that complex 3D scene generation is significantly challenging. However, the technical contributions of this work are fairly straightforward and not very inspiring. First, the accurate layout of 3D scene is largely due to the planning ability of LLM. The proposed program-aided planning does reduce the hallucination but is somewhat engineering. Second, semantic SDS directly utilizes semantic masks for SDS, which may be able to maintain the semantics from the initial layout, but the visual quality is not satisfactory. As shown in Figure 7, for "a house that is half made of LEGO and half of snow", the area where the attributes change is noisy.**
>
> A4: Acknowledging the intricate challenges posed by complex 3D scene generation, our approach strategically simplifies this complex task by decomposing it into distinct planning and generation phases. The utilization of Large Language Models (LLMs) as a proficient program-aided planning mechanism underscores our commitment to leveraging the most suitable tools available, with no current alternatives surpassing LLMs in effectiveness for this purpose. Concerning the visual quality, as discussed in A1: our methodology has set a new SOTA result in compositional generation quality across the models. While individual optimization of objects through tailored iterations and hyperparameters could potentially enhance results, such an approach diverges from our core objectives. More crucially, our SemanticSDS framework stands out for its versatility and applicability across a broad spectrum of existing generative models, thereby demonstrating the robustness and innovative capacity of our proposed solution.
>
>
> **Q5. Alternating training involves a tradeoff between local and global optimization. An analysis of this is important but not provided. For example, what are the results of "local optimization only" and "global optimization only"? And what is the influence of local-global ratio during alternating training?**
>
> A5: The alternating training method is employed to navigate the optimal balance between local and global optimizations effectively. As highlighted in the ablation study of GALA3D [1] (i.e., Figure 7 in GALA3D), it is evident that local optimization tends to enhance the visual generation quality of individual objects. In contrast, global optimization significantly improves the compositional harmony of multiple objects, ensuring their sizes, positions, and relationships are more accurately represented. The variation in the local-global optimization ratio inherently adjusts the focus of the optimization process, affecting the final outcome in terms of either individual detail fidelity or overall scene coherence. Considering this training technique is not our main contribution, thus we did not discuss it in our main paper. Nevertheless, we appreciate your insightful feedback on this matter.
>
> [1] Zhou, Xiaoyu, et al. "GALA3D: Towards Text-to-3D Complex Scene Generation via Layout-guided Generative Gaussian Splatting." ICML 2024.
>
> We hope our response helps you better understand our method and addresses your concerns. If you have any further questions, please feel free to reach out for further discussion.

---

### Author Response · Authors · 2024-11-22
**Global Response**

We sincerely thank all the reviewers for their thorough reviews and valuable feedback. We are glad to hear that our proposed framework is novel and effective (all reviewers), the design is well-motivated and notable (reviewer Pn2r and rjrF), and the performance improvements demonstrated in experiments are significant and promising, especially in attribute and spatial alignment (all reviewers). We have revised the manuscript according to the suggestions of reviewers (**marked in blue**).

We summarize our responses to the reviewers' comments as follows:

- We additionally provide more quantitative comparisons and visual synthesis results to demonstrate the superior generation quality of our framework.

- We conduct more ablation studies and provide more experimental details to validate the effectiveness of each design of our framework.

- We present an efficiency analysis and comparison to illustrate the practical feasibility of our work.

We reply to each reviewer's questions in detail below their reviews. Please kindly check them out. Thank you and please feel free to ask any further questions.

---

### Meta-Review · Area_Chair_tVtE · 2024-12-18

**Metareview:**

The paper in question introduces a novel mechanism, Semantic Score Distillation Sampling (SemanticSDS), for text-to-3D generation aimed at improving the compositional layout of objects in a 3D scene. The approach leverages program-aided layout planning and semantic maps for better control and precision. The authors posit that their methodology advances performance benchmarks and presents a unique semantic embedding strategy for achieving consistency in rendering views and distinguishing objects.

The strengths of this work lie in its innovative approach to a complex problem in the text-to-3D generation domain, with the reviewers acknowledging the promise shown by the experimental outcomes. The introduction of SemanticSDS is particularly noted for its potential in enhancing attribute and spatial alignments in 3D scenes.

However, the visual quality of the generated textures has been called into question due to apparent blurring and artifacts. The decision to reject the paper pivots on the unresolved issues surrounding the visual quality of the textures—crucial for the credibility of 3D generation—and the inadequate comparison with existing methodologies that could better contextualize the paper's contributions.

**Additional Comments On Reviewer Discussion:**

During the discussion, Reviewer Pn2r expressed concerns regarding the reproducibility of the results due to the visual quality issues, while Reviewer rjrF highlighted the need for more robust baselines for comparison.

The authors responded with additional visual results to address the quality concerns and appended further details and data in the appendix to clarify the comparisons and discuss failure cases. These efforts partially satisfied Reviewer rjrF and Reviewer hCfH, although the primary concern highlighted by Reviewer Pn2r regarding visual quality persisted post-rebuttal.

In weighting these concerns, the persistent issue of visual quality was given considerable importance due to its fundamental role in 3D generation. Although the authors' rebuttal addressed certain points, the core issue identified by Reviewer Pn2r remained, leading to the decision to reject.

---

### Decision · Program_Chairs · 2025-01-22

Reject